# TRACE: Your Diffusion Model is Secretly an Instance Edge Detector

**Sanghyun Jo**[1][*], **Ziseok Lee**[2][*], **Wooyeol Lee**[2],
**Jonghyun Choi**[2], **Jaesik Park**[2][†], **Kyungsu Kim**[2][†]
[1]OGQ, Seoul, Korea    [2]Seoul National University, Seoul, Korea

## Abstract

High-quality instance and panoptic segmentation has traditionally relied on dense instance-level annotations such as masks, boxes, or points, which are costly, inconsistent, and difficult to scale. Unsupervised and weakly-supervised approaches reduce this burden but remain constrained by semantic backbone constraints and human bias, often producing merged or fragmented outputs. We present TRACE (TRAnsforming diffusion Cues to instance Edges), showing that text-to-image diffusion models secretly function as instance edge annotators. TRACE identifies the Instance Emergence Point (IEP) where object boundaries first appear in self-attention maps, extracts boundaries through Attention Boundary Divergence (ABDiv), and distills them into a lightweight one-step edge decoder. This design removes the need for per-image diffusion inversion, achieving 81× faster inference while producing sharper and more connected boundaries. On the COCO benchmark, TRACE improves unsupervised instance segmentation by +5.1 AP, and in tag-supervised panoptic segmentation it outperforms point-supervised baselines by +1.7 PQ without using any instance-level labels. These results reveal that diffusion models encode hidden instance boundary priors, and that decoding these signals offers a practical and scalable alternative to costly manual annotation.
**Project Page:** shjo-april.github.io/TRACE

## 1 Introduction

Panoptic segmentation unifies semantic and instance segmentation and underpins real-world applications, such as autonomous driving (Elharrouss et al., 2021; Zendel et al., 2022). However, achieving reliable instance-level delineation has long relied on dense pixel-wise annotations such as masks, boxes, or points, which are prohibitively expensive, inconsistent across annotators, and fundamentally hard to scale. These limitations motivate the search for annotation-free alternatives that can retain the fine granularity of supervised methods without the cost of labeling.

Recent unsupervised and weakly-supervised approaches (Sick et al., 2025; Li et al., 2024) attempt to bypass dense labeling. Unsupervised instance segmentation (UIS) eliminates explicit annotation by clustering semantic features from pretrained vision transformers (Caron et al., 2021; Oquab et al., 2023), but these models are inherently optimized for semantic similarity across images rather than instance separation within an image. As shown in Fig. 2, existing UIS methods (Wang et al., 2023a; Li & Shin, 2024) often merge adjacent objects of the same class or fragment single instances, and rely on heuristic assumptions such as a predefined number of objects. Parallel advances in weakly-supervised semantic segmentation have demonstrated near-supervised performance, achieving up to 99% fully supervised accuracy on VOC 2012 (Everingham et al., 2010) using only image-level tags (Jo et al., 2023; 2024a). However, extending this success to the panoptic setup still requires point or box annotations to disambiguate instances. These additional annotations remain costly and error-prone, particularly when objects overlap or when annotators are inconsistent.

Our key insight is that *self-attention maps of text-to-image diffusion models (Podell et al., 2023; Esser et al., 2024) encode instance-aware cues early in denoising*. As shown in Fig. 1(a), cross-attention does not reliably separate adjacent objects even with an explicit prompt, whereas self-attention at specific timesteps reveals instance-level structure. During the denoising process, the model transitions from noise to instance-level structure and then to semantic content. This raises

---

[*]Equal contribution.    [†]Corresponding authors: {jaesik.park, kyskim}@snu.ac.kr

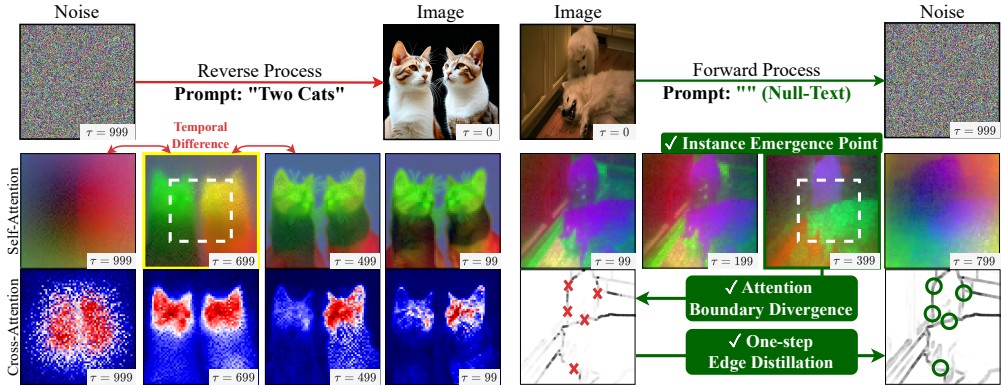

**(a) Observation**: Instance Cues in Diffusion Models    **(b) TRACE**: Extracting Instance Boundaries

Figure 1: **Emergence and extraction of instance cues in diffusion attention.** (a) In reverse process, cross-attention remains semantic even with the prompt, whereas self-attention at specific steps reveals instance-level structure. (b) TRACE selects the instance-emergent step using a temporal divergence criterion, extracts non-parametric edges from self-attention differences, and refines them via a one-step distillation with the diffusion backbone to refine instance boundaries.

a central question: can diffusion self-attention itself serve as an annotation-free source of instance-level edge maps?

To answer this, we introduce **TRACE** (**TRA**nsforming diffusion **C**ues to instance **E**dges), a framework that decodes instance boundaries directly from pretrained text-to-image diffusion models. As illustrated in Fig. 1(b), TRACE first identifies the Instance Emergence Point (IEP) by measuring temporal divergence to select the timestep where the instance structure is most pronounced. It then applies Attention Boundary Divergence (ABDiv) to score criss-cross differences in self-attention and generate initial edge maps. At this stage, pixels within the same object exhibit nearly identical self-attention distributions, whereas pixels across different objects diverge sharply; this divergence peaks on true instance boundaries and provides a direct signal for instance edge extraction. To reduce the computational cost of the per-image forward process at test time, these edges are distilled into a one-step predictor that integrates the diffusion backbone with an edge decoder. The resulting edges are used as boundary priors in downstream segmentation methods (Wang et al., 2023a; Jo et al., 2024a), guiding the propagation to cleanly separate adjacent objects by splitting merged regions along instance boundaries (Fig. 2).

Our key contributions are summarized as follows:
- We observe that self-attention in diffusion models briefly yet reliably reveals instance-level structure during denoising, unlike common vision transformers (see Tab. 5).
- The proposed TRACE unifies two key ideas for annotation-free instance boundary discovery: the Instance Emergence Point and Attention Boundary Divergence.
- TRACE enables annotation-free instance and panoptic segmentation: 1) Improves unsupervised instance segmentation baselines by +4.4 AP with only 6% runtime overhead; 2) With tag supervision, surpasses point-supervised panoptic models, up to +7.1 PQ on VOC 2012; 3) As seeds for SAM, outperforms open-vocabulary detectors by up to +16.5 PQ (stuff).

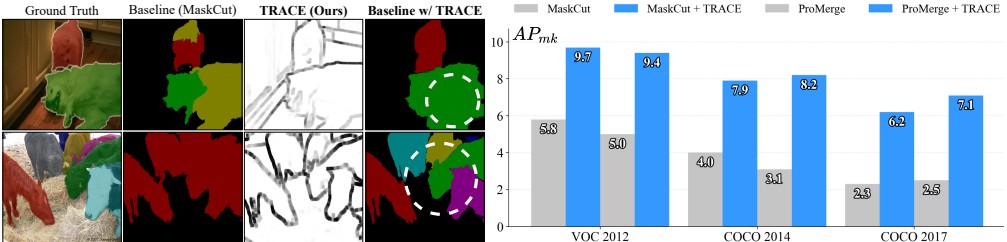

Figure 2: **Effect of TRACE.** (*Left*) Our instance edges decoded from diffusion self-attention for reconnection of fragmented masks and separation of adjacent objects, with white dotted circles marking corrected boundaries. (*Right*) Consistent $AP_{mk}$ gains over baselines (Wang et al., 2023a; Li & Shin, 2024) without instance-level annotations.

## 2 RELATED WORK

**Unsupervised Instance Segmentation.** Instance segmentation aims to delineate individual objects and typically requires pixel-level annotations. Early methods (Wang et al., 2022; Ishtiak et al., 2023) learn pseudo masks from external features (Wang et al., 2021) but require training from scratch and show limited accuracy. Recent approaches, such as MaskCut (Wang et al., 2023a), U2Seg (Niu et al., 2024), and UnSAM (Wang et al., 2024b), cluster features from pretrained vision transformers like DINO (Caron et al., 2021). These models strong at semantic grouping across images but are not explicitly designed for separating instances within an image. Therefore, these clustering methods often rely on heuristics such as a maximum number of instances or confidence thresholds and tend to merge adjacent objects of the same category.

To improve instance separation, CutS3D (Sick et al., 2025) and CUPS (Hahn et al., 2025) incorporate monocular depth estimators (Ke et al., 2024; Yang et al., 2024) to split objects at different ranges. However, depth-based approaches struggle when neighboring objects lie at similar depth and they degrade on distant or small objects where estimated depth becomes blurry. By contrast, our diffusion-based strategy (TRACE) extracts instance boundaries from self-attention of pretrained diffusion models (Peebles & Xie, 2023; Bao et al., 2023; Podell et al., 2023; Esser et al., 2024). This boundary-centric cue does not assume the number of instances, is robust to object scale and distance, and refines existing unsupervised pipelines (Wang et al., 2023a; Li & Shin, 2024) without any supervision or retraining, achieving up to 29.1% higher performance on COCO compared to depth-based methods (see Tab. 1).

**Weakly-supervised Semantic and Panoptic Segmentation.** Panoptic segmentation jointly requires semantic masks for "stuff" regions (*e.g.*, grass) and instance masks for "thing" objects (*e.g.*, person), which makes it one of the most annotation-intensive tasks in segmentation. To reduce this labeling cost, weakly-supervised panoptic segmentation have been explored. Image-level class tags (Shen et al., 2021) alone cannot separate instances. Bounding boxes are ill-suited for panoptic supervision because they provide only coarse rectangles for "thing" objects and cannot define the non-overlapping pixel-wise regions required for "stuff" regions. Consequently, point annotations (Fan et al., 2022; Li et al., 2023b; 2024) have become the dominant form of weak supervision. However, points vary across annotators and are often placed near object centers, which produces partial or missed instances and leaves adjacent objects merged (see Fig. 3).

Meanwhile, in weakly-supervised semantic segmentation, recent tag-supervised approaches (Jo et al., 2024a; Yang et al., 2025a;b) show that image tags alone can reach about 95% of fully supervised accuracy on the PASCAL VOC 2012 benchmark, indicating that tags are sufficient for semantics but not for instance separation. Therefore, we revisit tag supervision and inject instance structure using diffusion priors: TRACE attaches to a tag-supervised semantic model (Jo et al., 2023; 2024a) and converts its pseudo semantic masks into pseudo panoptic masks by supplying instance-aware boundaries from diffusion self-attention. This model-agnostic design uses only image-level tags, cleanly separates adjacent objects, and first surpasses point-supervised panoptic baselines (Li et al., 2024) on VOC and COCO benchmarks (see Tab. 2).

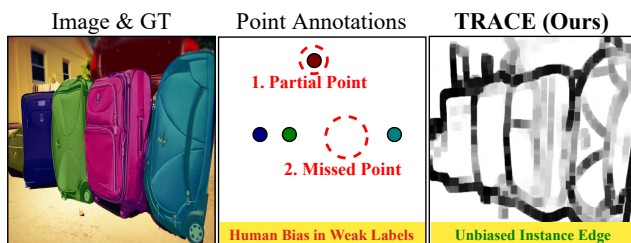

Figure 3: Example of human bias in COCO.

**Diffusion-Driven and Open-Vocabulary Segmentation.** Recent approaches, including DiffCut (Couairon et al., 2024), DiffSeg (Tian et al., 2024), and ConceptAttention (Helbling et al., 2025), repurpose the self- and cross-attention maps of pretrained text-to-image diffusion models (Peebles & Xie, 2023; Bao et al., 2023; Esser et al., 2024; Podell et al., 2023) for semantic segmentation by analyzing attention at a fixed timestep without inversion. In parallel, open-vocabulary segmentation builds on contrastive pretraining (Radford et al., 2021) to map free-form text to visual concepts, enabling text-conditioned masks. Despite progress, such models (Liu et al., 2024b; You et al., 2023; Zhao et al., 2025) typically underperform closed-vocabulary segmentation and struggle to produce reliable seeds under multi-tag inputs because they are hard to obtain from captions with limited tag coverage or in scenes containing multiple nearby objects. Compared to them, TRACE yields higher panoptic quality than open-vocabulary detection when used as TRACE seeds for SAM in Sec. 6.

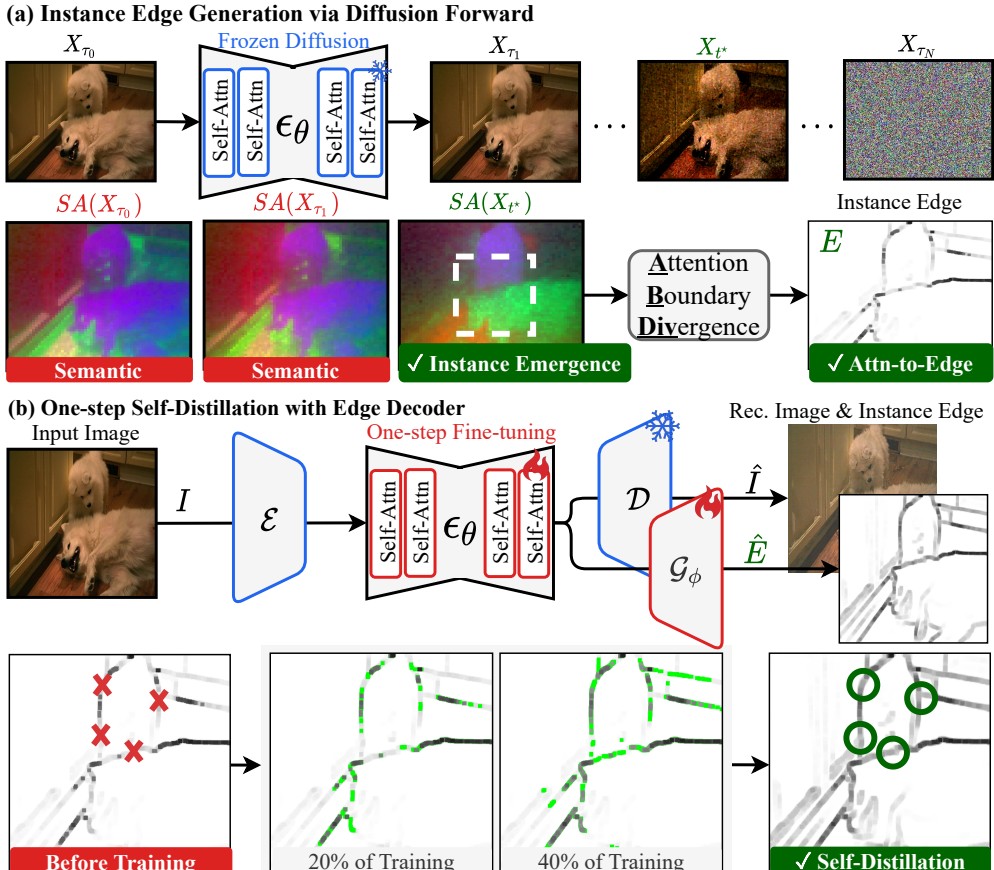

Figure 4: **Overview of TRACE.** (a) Diffusion forward locates the instance emergence point $t^\star$ (IEP) via a KL peak and extracts the instance-aware attention $SA(X_{t^\star})$; ABDiv converts it into a pseudo edge map $E$. (b) One step self distillation at $t=0$ trains an edge decoder $\mathcal{G}_\phi$ with $E$, masking uncertain pixels. Training from $E$ closes gaps in fragmented edges (green circles) and yields connected boundaries $\hat{E}$. At inference, TRACE predicts $\hat{E}$ in a single pass w/o IEP or ABDiv.

## 3 METHOD

In this section, we outline an overview of TRACE in Fig. 4 for a comprehensive understanding of our framework. Section 3.2 introduces the Instance Emergence Point (IEP), which selects the denoising step where instance structure is most pronounced. Section 3.3 describes Attention Boundary Divergence (ABDiv), which converts criss-cross self-attention differences into a pseudo edge map. Section 3.4 details a one-step distillation that trains an edge decoder with the diffusion backbone to produce connected edges and enable real-time inference. Section 3.5 shows how our edges integrate with downstream segmentation models through Background-Guided Propagation (BGP).

### 3.1 BACKGROUND

**Models.** Diffusion (Rombach et al., 2022; Podell et al., 2023) and flow matching models (Esser et al., 2024; Lipman et al., 2022) generate images by learning a reverse process from noise to data. Although their objectives differ, both families use similar transformer backbones with self- and cross-attention. We refer to either as a diffusion model since our method is compatible with both. In typical text-to-image implementations, a VAE encoder $\mathcal{E}$ maps an input image $I$ into a latent $X_0 = \mathcal{E}(I)$, a denoising network $\epsilon_\theta$ iteratively predicts noise (or velocity) on latents $X_t$ conditioned on an optional text embedding $c$, and a VAE decoder $\mathcal{D}$ turns a generated image $\hat{I} = \mathcal{D}(\hat{X}_0)$. TRACE reads only self-attention maps from a diffusion model and does not require text prompts.

**Self-Attention Collection and Aggregation.** For a latent $X_t \in \mathbb{R}^{HW \times d}$ at step $t$, we form queries $Q_t = X_t W_Q$ and keys $K_t = X_t W_K$. The self-attention for block $k$ (averaged over heads, rows sum to 1) is $SA_t^k(X_t) = \mathrm{softmax}(Q_t K_t^\top / \sqrt{d}) \in [0, 1]^{HW \times HW}$; we ignore cross-attention and do

not provide prompts. To fuse maps from blocks $k = 1, \ldots, N$ that may operate at different spatial sizes $w_k$ (*e.g.*, multi-scale stages in U-Net; for single-scale DiT-style backbones $\mathcal{U}$ is identity), we upsample to $w_{\max}$ and average: $SA(X_t) = \frac{1}{N} \sum_{k=1}^{N} \mathcal{U}_{k \to \max}(SA_t^k(X_t))$. We implement this with PyTorch forward hooks $\mathcal{H}$ on attention blocks; on each forward of the denoising network $\epsilon_\theta$ with inputs $(X_0, t)$, the hook collects $\{SA_t^k\}$, performs the aggregation, and returns the map, $\mathcal{H}(\epsilon_\theta, X_0, t) = SA(X_t)$, without altering model outputs.

## 3.2 IDENTIFYING THE INSTANCE-AWARE DENOISING STEP

We next ask: *at which point of the denoising trajectory does self-attention truly become instance-aware?* Early in denoising, self-attention maps are almost indistinguishable from noise. As steps proceed, we observe a sharp rise in the Kullback–Leibler (KL) divergence between consecutive maps. This peak coincides with the emergence of clear object boundaries, after which divergence gradually decreases as object shape stabilizes while semantics continue to refine. During inversion, the trajectory unfolds in reverse order: semantic $\to$ instance $\to$ noise.

Motivated by our observation, we propose the Instance Emergence Point (IEP) as the timestep $t^\star$ where this divergence is maximized:

$$t^\star = \underset{t \in \{\tau_1, \ldots, \tau_N\}}{\operatorname{argmax}} D_{\mathrm{KL}}(SA(X_{t_{\mathrm{prev}}}) \| SA(X_t)) \tag{1}$$

where $\tau_0 < \cdots < \tau_N$ are discrete timesteps (with $t = \tau_k$, $t_{\mathrm{prev}} = \tau_{k-1}$). The self-attention map at this point, $SA(X_{t^\star})$, is denoted as $SA_{\mathrm{inst}}$ and serves as our instance-aware representation. Specifically, KL divergence is a natural choice (Tian et al., 2024) because each row of $SA(X_t)$ is a probability distribution. Unlike mean-squared or absolute differences, KL's log-scale sensitivity amplifies subtle but meaningful variations in high-dimensional self-attention that directly align with boundary emergence (see Tab. 4). In practice, we adopt a fixed inversion stride of 100 steps for efficiency and accuracy. A detailed analysis of step size and the distribution of the optimal timestep $t^\star$ across diffusion backbones is provided in Fig. 7.

## 3.3 EXTRACTING INSTANCE EDGES FROM SELF-ATTENTION

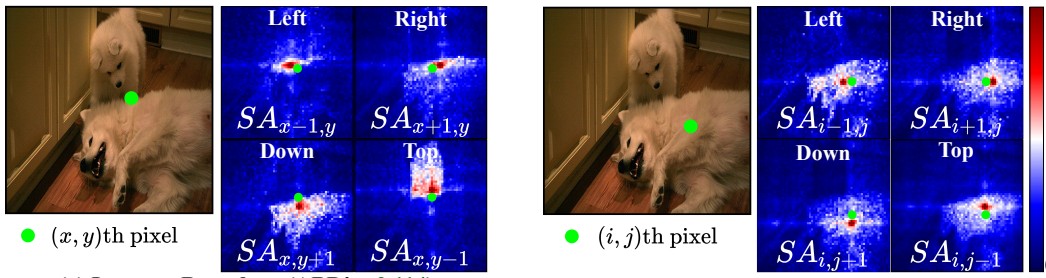

(a) Instance Boundary (ABDiv: 0.114)  (b) Instance Interior (ABDiv: 0.027)

Figure 5: **Illustration of Attention Boundary Divergence (ABDiv).** Boundary regions (a) exhibit sharp attention divergence between opposite neighbors, whereas interior regions (b) remain stable, producing much smaller ABDiv values.

Neighboring pixels within the same instance exhibit similar self-attention maps, whereas those across instance boundaries differ, as shown in Fig. 5. We convert this contrast into edges with Attention Boundary Divergence (ABDiv), a simple non-parametric score that transforms instance-aware self-attention maps into boundary maps without clustering or annotations. We apply ABDiv on the instance-aware map $SA_{\mathrm{inst}} = SA(X_{t^\star})$ identified in Sec. 3.2. For a pixel $(i, j)$, ABDiv aggregates the divergence between opposite 4-neighbors:

$$\mathrm{ABDiv}(SA)_{i,j} := D_{\mathrm{KL}}(SA_{i+1,j} \| SA_{i-1,j}) + D_{\mathrm{KL}}(SA_{i,j+1} \| SA_{i,j-1}). \tag{2}$$

By default, ABDiv is computed with a 4-neighborhood using opposite pairs (left/right and top/bottom) as defined in Eq. 2. An 8-neighborhood extension that adds diagonal pairs achieves the same accuracy while increasing computation by approximately $2\times$, so we adopt the 4-neighborhood in all experiments.

### 3.4 ONE-STEP SELF-DISTILLATION WITH EDGE DECODER

Starting from the instance-aware self-attention map $SA_{\text{inst}} = SA(X_{t^\star})$ identified in Sec. 3.2, we obtain an initial edge map $E$ by applying ABDiv as defined in Eq. 2. Inspired by pseudo-labeling strategies for weak supervision (Ahn et al., 2019; Jo et al., 2024a), we adopt a reliability-based thresholding to mitigate label noise from ambiguous self-attention signals. Specifically, we define a ternary map using the mean $\mu$ and standard deviation $\sigma$ of ABDiv scores: pixels $> \mu + \sigma$ are edges $1$, $< \mu - \sigma$ are interior $0$, and the intermediate range is marked as uncertain $-1$. These uncertain pixels are explicitly excluded from the loss computation, which effectively suppresses false positives while maintaining high recall (see ablation in Appendix E.1 and Tab. 10).

To replace a per-image IEP+ABDiv computation at inference with a single pass, we fine-tune the diffusion backbone using Low-Rank Adaptation (Hu et al., 2022) and jointly train an edge decoder $\mathcal{G}_\phi$. Beyond efficiency, the decoder also learns to complete fragmented edges, following common practice in boundary detection (Xie & Tu, 2015; Xiao et al., 2018; Su et al., 2023). Let $I, \hat{I} \in \mathbb{R}^{H \times W \times 3}$ be the original and reconstructed images, $E, \hat{E} = \mathcal{G}_\phi(\mathcal{H}(\epsilon_\theta, I, t=0)) \in [-1, 1]^{H \times W}$ be the pseudo edge map of ABDiv and the edge predicted by $\mathcal{G}_\phi$. Our training objective is $\mathcal{L}(\theta, \phi) = \mathcal{L}_{\text{rec}}(\theta) + \mathcal{L}_{\text{edge}}(\theta, \phi) = \|I - \hat{I}\|^2 + \text{DiceLoss}(E, \hat{E})$. After training, a single forward pass at $t{=}0$ produces a connected and precise edge map and removes the need for IEP and ABDiv during inference, as shown in Fig. 4. Full algorithmic details are provided in Appendix C.

### 3.5 SEMANTIC-TO-INSTANCE MASK REFINEMENT WITH INSTANCE EDGES

We now use our instance edges to regularize and complete segmentation masks. Given segmentation masks, connected component labeling treats the TRACE edges as separators and assigns unique labels to connected regions that are not cut by edges. Inspired by (Ahn et al., 2019), we design the Background-Guided Propagation (BGP), as shown in Fig. 6, propagate each fragmented mask inside its instance boundaries to close gaps and produce smooth regions. We then iteratively merge overlapping masks whose intersection over union exceeds $\tau_{\text{BGP}} = 0.5$ until convergence. This produces complete instance masks with our edges. We empirically find that performance remains stable over typical choices of $\tau_{\text{BGP}}$ on VOC, so we simply use $0.5$ in all experiments.

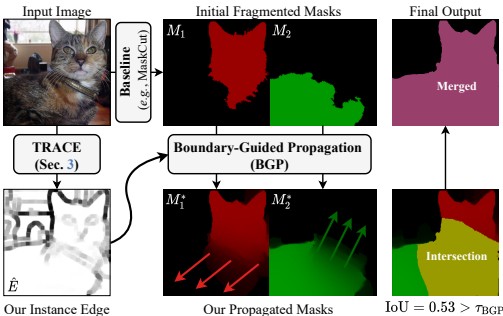

Figure 6: **Illustration of Boundary-Guided Propagation.** Fragmented masks spread within instance edges, and intersections (yellow) are resolved by edge respecting merging.

## 4 EXPERIMENTS

**Implementation Details.** For fair comparison, we follow standard protocols (Wang et al., 2023a; Li et al., 2023b) and run all experiments on a single NVIDIA A100 GPU. Stable Diffusion 3.5 Large (SD3.5-L) (Esser et al., 2024), our default backbone, performs best overall among five diffusion backbones evaluated (Tab. 5), with VOC and COCO as the main benchmarks and five additional datasets reported in Appendix E. Training details and evaluation metrics appear in Appendix C.

**Unsupervised Instance Segmentation.** In Tab. 1, TRACE refines masks produced by existing UIS methods Wang et al. (2023a); Li & Shin (2024); Wang et al. (2024b) and consistently improves performance $AP^{mk}$ with gains ranging from +3.6 to +5.3 points. For clarity, we group results into training-free and fine-tuned methods, where the latter relies on a Mask R-CNN He et al. (2017) trained on pseudo instance masks. In particular, compared to the depth prior (Sick et al., 2025), TRACE attains higher $AP^{mk}$ on COCO (+2.2/+2.1 on 2014/2017), highlighting the advantage of diffusion-driven instance edges. Qualitative results appear in Fig. 17.

**Weakly-supervised Panoptic Segmentation.** We refine semantic masks from tag-supervised methods (Jo et al., 2023; 2024a) with instance-aware edges to form pseudo panoptic masks and then train a standard Mask2Former (Cheng et al., 2022) following the common evaluation protocol. In particular, DHR (Jo et al., 2024a) with TRACE surpasses point-supervised counterparts (Li et al., 2023b; 2024) by using only image-level tags (see Tab. 2 and Fig. 3), indicating that our edges provide the instance geometry that tag supervision lacks. Qualitative examples are provided in Fig. 16.

Table 1: Performance of unsupervised instance segmentation.

(a) Training-free UIS

| Method | VOC 2012 | | COCO 2014 | | COCO 2017 | |
|---|---|---|---|---|---|---|
| | $AP^{mk}$ | $AR^{mk}_{100}$ | $AP^{mk}$ | $AR^{mk}_{100}$ | $AP^{mk}$ | $AR^{mk}_{100}$ |
| MaskCut* | 5.8 | 14.0 | 3.0 | 6.7 | 2.3 | 6.5 |
| + UnSAM* | 6.1 | 14.5 | 3.3 | 6.9 | 2.5 | 6.8 |
| + TRACE (Ours) | **9.7** | **18.4** | **7.9** | **12.6** | **7.5** | **9.8** |
| ProMerge* | 5.0 | 13.9 | 3.1 | 7.6 | 2.5 | 7.5 |
| + TRACE (Ours) | **9.4** | **18.2** | **8.2** | **13.1** | **7.8** | **11.2** |

(b) Fine-tuned UIS

| Method | VOC 2012 | | COCO 2014 | | COCO 2017 | |
|---|---|---|---|---|---|---|
| | $AP^{mk}$ | $AR^{mk}_{100}$ | $AP^{mk}$ | $AR^{mk}_{100}$ | $AP^{mk}$ | $AR^{mk}_{100}$ |
| CutLER* | 11.2 | 34.2 | 8.9 | 25.1 | 8.7 | 24.9 |
| + CutS3D | - | - | 10.9 | - | 10.7 | - |
| + TRACE (Ours) | **14.8** | **45.3** | **13.1** | **34.6** | **12.8** | **33.9** |
| ProMerge+* | 11.1 | 33.1 | 9.0 | 25.3 | 8.9 | 25.1 |
| + TRACE (Ours) | **15.0** | **43.8** | **13.3** | **35.1** | **13.0** | **25.8** |

Table 2: Performance of weakly-supervised panoptic segmentation.

| Method | Backbone | Supervision | VOC 2012 | | | | | COCO 2017 | | | | |
|---|---|---|---|---|---|---|---|---|---|---|---|---|
| | | | PQ | $PQ^{th}$ | $PQ^{st}$ | SQ | RQ | PQ | $PQ^{th}$ | $PQ^{st}$ | SQ | RQ |
| Panoptic FCN (Li et al., 2021) | ResNet-50 | $\mathcal{M}$ | 67.9 | 66.6 | 92.9 | - | - | 43.6 | 49.3 | 35.0 | 80.6 | 52.6 |
| Mask2Former* (Cheng et al., 2022) | ResNet-50 | $\mathcal{M}$ | 73.6 | 72.6 | 93.5 | 90.6 | 80.5 | 51.9 | 57.7 | 43.0 | - | - |
| PSPS (Fan et al., 2022) | ResNet-50 | $\mathcal{P}$ | 49.8 | 47.8 | 89.5 | - | - | 29.3 | 29.3 | 29.4 | - | - |
| Panoptic FCN (Li et al., 2021) | ResNet-50 | $\mathcal{P}$ | 48.0 | 46.2 | 85.2 | - | - | 31.2 | 35.7 | 24.3 | - | - |
| Point2Mask* (Li et al., 2023b) | ResNet-50 | $\mathcal{P}$ | 53.8 | 51.9 | 90.5 | - | - | 32.4 | 32.6 | 32.2 | 75.1 | 41.5 |
| EPLD (Li et al., 2024) | ResNet-50 | $\mathcal{P}$ | 56.6 | 54.9 | 89.6 | - | - | 34.2 | 33.6 | 35.3 | - | - |
| Point2Mask* (Li et al., 2023b) | Swin-L | $\mathcal{P}$ | 61.0 | 59.4 | 93.0 | - | - | 37.0 | 37.0 | 36.9 | 75.8 | 47.2 |
| EPLD (Li et al., 2024) | Swin-L | $\mathcal{P}$ | 68.5 | 67.3 | 93.4 | - | - | 41.0 | 39.9 | 42.7 | - | - |
| JTSM (Shen et al., 2021) | ResNet-18-WS | $\mathcal{I}$ | 39.0 | 37.1 | 77.7 | - | - | 5.3 | 8.4 | 0.7 | 30.8 | 7.8 |
| MARS* (Jo et al., 2023) | ResNet-50 | $\mathcal{I}$ | 41.4 | 39.8 | 85.3 | 83.0 | 57.8 | 11.7 | 13.3 | 10.2 | 58.3 | 11.8 |
| + TRACE (Ours) | ResNet-50 | $\mathcal{I}$ | 50.4 | 48.5 | 88.9 | 86.6 | 60.1 | 29.5 | 31.1 | 28.9 | 62.5 | 39.3 |
| DHR* (Jo et al., 2024a) | ResNet-50 | $\mathcal{I}$ | 45.0 | 43.3 | 88.3 | 83.3 | 59.8 | 18.3 | 17.5 | 18.1 | 69.3 | 14.8 |
| + TRACE (Ours) | ResNet-50 | $\mathcal{I}$ | 56.9 | 55.2 | 91.0 | 88.4 | 63.4 | 32.8 | 32.7 | 32.9 | 75.5 | 42.5 |
| + TRACE (Ours) | Swin-L | $\mathcal{I}$ | **69.8** | **68.4** | **96.2** | **94.5** | **71.2** | **43.1** | **42.5** | **43.5** | **83.8** | **55.3** |

$\mathcal{M}$: Full mask supervision (upper bound), $\mathcal{P}$: One point per instance, $\mathcal{I}$: Image-level tags only (no instance annotations)
* Reproduced results using the publicly accessible code. The rest are the values reported in the publication.

## 5 ABLATION STUDY

**Component Ablation.** Table 3 shows how each stage steers TRACE from purely semantic cues toward instance delineation. Because the Instance Emergence Point (IEP) marks the timestep where semantic attention first becomes instance-aware, it cannot be evaluated on its own: without the boundary scoring of ABDiv there is no measurable edge signal. Accordingly, case (b) applies Attention Boundary Divergence (ABDiv) at a purely semantic

Table 3: Effect of key components on COCO 2014 with the UIS baseline (Li & Shin, 2024).

| | IEP (Sec. 3.2) | ABDiv (Sec. 3.3) | Distill (Sec. 3.4) | $AP^{mk}$ |
|---|---|---|---|---|
| (a) | ✗ | ✗ | ✗ | 3.1 |
| (b) | ✗ | ✓ | ✗ | 3.2 |
| (c) | ✓ | ✓ | ✗ | 4.8 |
| (d) | ✓ | ✓ | ✓ | **8.2** |

timestep, following prior diffusion approaches (Tian et al., 2024; Couairon et al., 2024), and yields almost no gain, confirming that semantic self-attention alone cannot reveal instance edges. Introducing IEP in case (c) pinpoints the denoising step where diffusion self-attention transitions from semantic grouping to instance structure, enabling ABDiv to capture instance boundaries. Finally, case (d) adds one-step self-distillation to compress these transient cues into a single-pass predictor, eliminating per-image IEP and ABDiv at inference and cutting latency from 3682 ms to just 45 ms per image (about 81× faster) while preserving and even strengthening edge connectivity.

**Self-Distillation with Reconstruction.** During one-step self-distillation (Sec. 3.4, we jointly optimize edge prediction and image reconstruction. While the edge loss $\mathcal{L}_{edge}$ compels the student to reproduce the teacher's instance boundaries, edges alone can overfit to noisy or incomplete supervision. Adding a reconstruction loss $\mathcal{L}_{rec}$ anchors the decoder to global image structure, stabilizing training and suppressing artifacts along low-contrast boundaries. This auxiliary objective yields smoother and more coherent edges and provides measurable gains in both accuracy and perceptual quality ($AP^{mk}$ from 8.9 to 9.4; SSIM from 0.71 to 0.83) without adding inference cost.

**Instance Emergence Analysis.** Figure 7 evaluates the Instance Emergence Point (IEP) along two axes. In Fig. 7(a), enlarging the denoising step size reduces the number of self-attention accumulation and thus latency, with negligible loss in $AP^{mk}$. A step size of 100 strikes the best balance, maintaining about 9.4 $AP^{mk}$ with roughly 3 s of IEP search per image. In Fig. 7(b), the optimal timestep $t^{\star}$ clusters tightly across five diffusion backbones, indicating a model-agnostic semantic-to-instance transition and supporting a fixed step size without per-image or per-model tuning. Extensive results in Appendix E.2 further confirm the robustness of IEP, demonstrating consistent $t^{\star}$

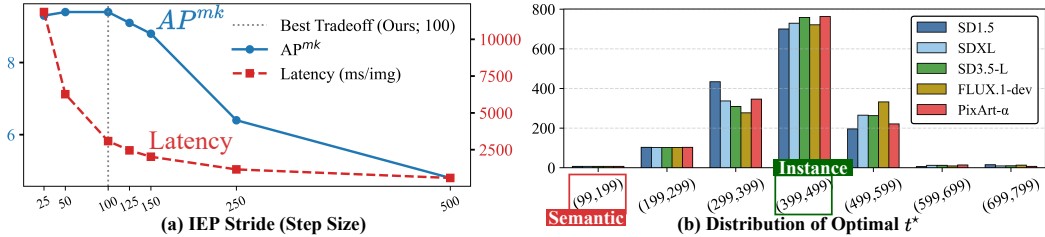

Figure 7: **Analysis of IEP.** (a) Accuracy–latency trade-off across different IEP step sizes. (b) Distribution of optimal timestep $t^\star$ showing a consistent semantic-to-instance transition.

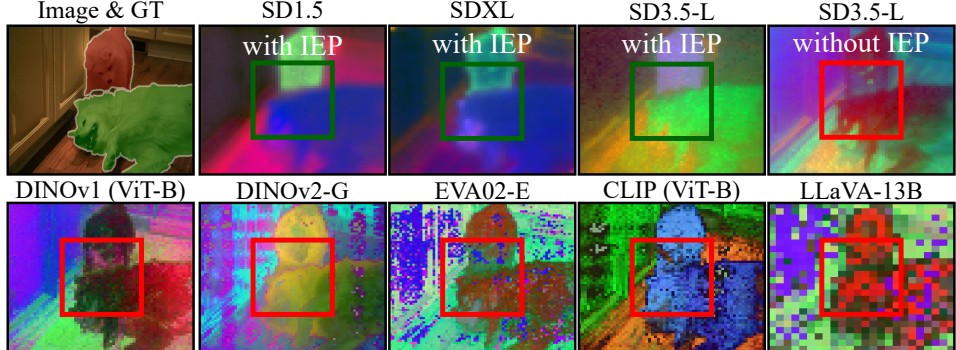

Figure 8: Visualization of self-attention maps with PCA.

distributions across datasets and classes, as well as the empirical superiority of our KL criterion over alternative metrics (*e.g.*, Entropy and Wasserstein).

**KL vs. Other Metrics.** To ensure our similarity choice is principled, we compared the Kullback–Leibler (KL) divergence with alternative metrics for both IEP and ABDiv. In Tab. 4, Jensen–Shannon divergence (JSD) achieves the similar $AP^{mk}$ but requires computing two KL terms against a mixture distribution, increasing latency by more than 60%. Mean-squared (L2) and mean-absolute (L1) losses reduce computation but sharply degrade accuracy, confirming KL as the most effective balance of precision and efficiency.

Table 4: Similarity metrics for IEP and ABDiv.

| IEP Metric | Latency/img | $AP^{mk}$ | ABDiv Metric | Latency/img | $AP^{mk}$ |
|---|---|---|---|---|---|
| KL (Ours) | 3,082 ms | 9.4 | KL (Ours) | 600 ms | 9.4 |
| JSD | 5,120 ms | 9.4 | JSD | 980 ms | 9.2 |
| MSE (L2) | 1,232 ms | 3.8 | MSE (L2) | 425 ms | 6.8 |
| MAE (L1) | 924 ms | 3.5 | MAE (L1) | 412 ms | 6.7 |

## 6 DISCUSSION

**Superiority of Generative Diffusion Priors.** To verify whether instance-aware cues are specific to diffusion models, we evaluate TRACE across 10 different backbones, including 5 diffusion and 5 non-diffusion foundation models (Oquab et al., 2023; Fang et al., 2024; Liu et al., 2024a; Siméoni et al., 2025; Podell et al., 2023) (see Tab. 5). Note that for non-diffusion backbones lacking temporal trajectories, we apply ABDiv (Sec. 3.3) directly to their self-attention maps. Remarkably, even the smallest diffusion model, PixArt-$\alpha$ (0.6B) (Chen et al., 2024), achieves 7.1 $AP^{mk}$, significantly outperforming the massive 72B-parameter Qwen2.5-VL (Bai et al., 2025) (4.1 $AP^{mk}$). This confirms that TRACE leverages the unique generative nature of diffusion models, where instance boundaries emerge during denoising (IEP; Sec. 3.2), rather than typical semantic features found in discriminative or multimodal models. Figure 8 visualizes this distinction: diffusion self-attention tightens along object boundaries at IEP, whereas non-diffusion attention collapses into coarse semantic blobs, failing to separate

Table 5: **Diffusion vs. Non-Diffusion.** TRACE fine-tuning results on COCO 2014. Blue rows indicate diffusion backbones.

| Method | Backbone | Params | $AP^{mk}$ | $AR^{mk}_{100}$ |
|---|---|---|---|---|
| ProMerge | – | – | 3.1 | 7.6 |
| *Non-diffusion backbones (ABDiv only)* | | | | |
| + TRACE | DINOv2-G | 1.1B | 2.6 | 7.7 |
| + TRACE | EVA02-E | 5.0B | 3.2 | 7.9 |
| + TRACE | DINOv3 | 7.0B | 4.3 | 8.9 |
| + TRACE | LLaVA | 13B | 3.8 | 8.4 |
| + TRACE | Qwen2.5-VL | 72B | 4.1 | 8.5 |
| *Diffusion backbones (IEP + ABDiv)* | | | | |
| + TRACE | SD1.5 | 0.8B | 6.8 | 11.2 |
| + TRACE | PixArt-$\alpha$ | **0.6B** | 7.1 | 11.8 |
| + TRACE | SDXL | 2.5B | 7.4 | 12.3 |
| + TRACE | SD3.5-L | 8.1B | 8.2 | 13.1 |
| + TRACE | FLUX.1 | 12B | **8.3** | **13.4** |

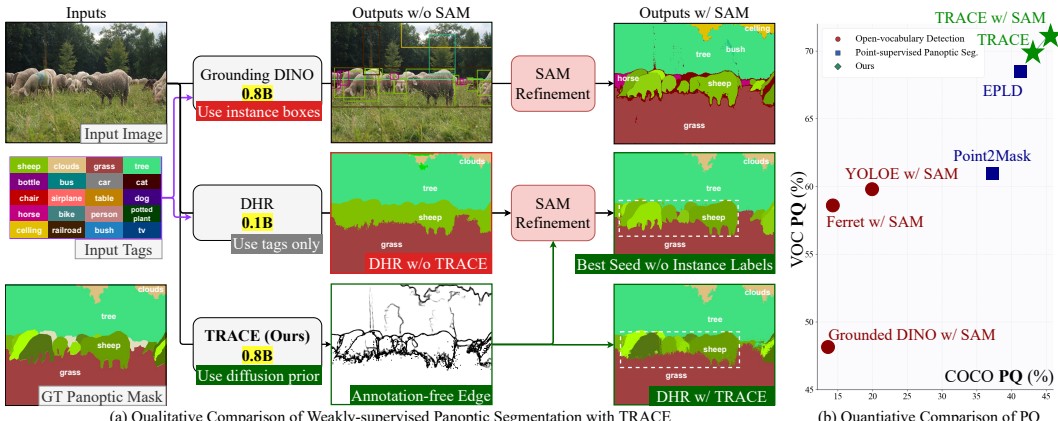

(a) Qualitative Comparison of Weakly-supervised Panoptic Segmentation with TRACE

(b) Quantiative Comparison of PQ

Figure 9: **Importance of TRACE.** (a) White dotted boxes mark regions where tag-supervised semantic masks are converted into panoptic masks by TRACE, cleanly separating adjacent instances. (b) Quantitative results show TRACE+SAM outperforming open-vocabulary detectors and TRACE+WSS (*e.g.*, DHR) surpassing point-supervised baselines.

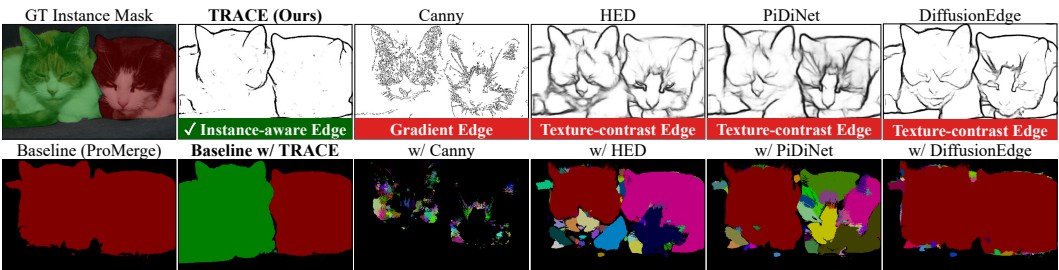

Figure 10: Instance-aware edge comparison with existing edge alternatives.

adjacent instances. Furthermore, within the diffusion family, performance correlates positively with model capacity (SD1.5 → FLUX.1), demonstrating that TRACE effectively scales with stronger generative priors while remaining model-agnostic in its applicability.

**Why Annotation-Free Instance Edges?** Recent efforts toward instance and panoptic segmentation often combine open-vocabulary detectors (Liu et al., 2024b; You et al., 2023; Zhao et al., 2025) with SAM (Kirillov et al., 2023), where the detector supplies instance-level boxes and SAM refines them into masks. Despite this progress, such pipelines still depend on box annotations and struggle when scenes involve many adjacent objects or ambiguous text prompts. TRACE provides a different alternative: it extracts instance edges directly from diffusion self-attention, requiring no instance supervision while offering clean separation of objects (Fig. 9). These edges are complementary to SAM, since SAM excels at refining seeds into precise instance masks while TRACE provides those seeds. This combination exceeds all supervised open-vocabulary baselines. In addition, when integrated with tag-supervised semantic models, TRACE supplies the missing instance geometry and converts their outputs into complete panoptic masks, outperforming point-supervised methods on VOC and COCO.

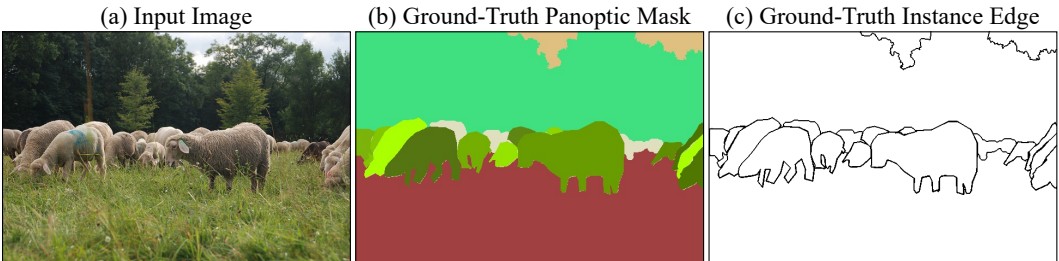

(a) Input Image  (b) Ground-Truth Panoptic Mask  (c) Ground-Truth Instance Edge

Figure 11: **Example of ground-truth instance boundary generation.** (a) Input image. (b) Ground-truth panoptic mask. (c) Ground-truth instance boundaries extracted from (b).

**Limitations of Conventional Edge Detectors.** Figure 10 tests whether conventional edges can replace TRACE for instance segmentation by swapping each detector's edge map into our BGP pipeline (Sec. 3.5). $AP^{mk}$ falls from 9.4 (TRACE) to 1.2 with Canny (Canny, 1986), 3.7 with HED (Xie & Tu, 2015), 4.1 with PiDiNet (Su et al., 2023), and 4.3 with DiffusionEdge (Ye et al., 2024). The gap reflects an objective mismatch: these methods are trained to predict RGB intensity-change contours (not instance edges), which makes them sensitive to texture and illumination.

**Quantitative Evaluation of Instance Boundaries.** Standard edge detection benchmarks (*e.g.*, BSDS (Arbelaez et al., 2010)) prioritize low-level texture or color contrast, which is misaligned with the goal of instance boundary detection. To evaluate instance-aware boundary quality, we construct a new benchmark from COCO 2014 panoptic masks, defining pixels between distinct segments as ground-truth edges (see Fig. 11). To assess quality, we report ODS and OIS metrics (Xie & Tu, 2015; Su et al., 2023) for boundary precision and clDice (Shit et al., 2021) for topological connectivity (see details in Appendix F). Table 6 compares TRACE against

Table 6: **Instance Edge Quality.** Evaluation on COCO 2014 validation set against ground-truth instance boundaries.

| Method | ODS | OIS | clDice |
|---|---|---|---|
| Canny | 0.129 | 0.202 | 0.134 |
| HED | 0.347 | 0.443 | 0.446 |
| PiDiNet | 0.362 | 0.450 | 0.574 |
| DiffusionEdge | 0.428 | 0.485 | 0.576 |
| **TRACE (Ours)** | **0.889** | **0.899** | **0.826** |

representative edge detectors (Canny, 1986; Xie & Tu, 2015; Su et al., 2023; Ye et al., 2024). TRACE achieves an ODS of **0.889**, more than doubling the performance of the strongest baseline (DiffusionEdge, 0.428). Conventional methods suffer from high false positives due to their sensitivity to internal textures, resulting in low precision for instance delineation. In contrast, TRACE effectively suppresses non-boundary gradients by leveraging diffusion priors. Furthermore, the superior clDice score (**0.826**) confirms that TRACE produces topologically connected boundaries, which is a critical property for successfully separating adjacent instances in downstream segmentation tasks.

**Limitations.** While TRACE demonstrates consistent improvements across 11 real-world benchmarks, including autonomous driving (see results in Tabs. 1, 2, 12, and 13), we identify limitations in specialized domains. First, for tiny instances ($\approx 0.01\%$ area) in satellite imagery (Wei et al., 2020; Waqas Zamir et al., 2019), performance degrades due to the spatial compression of the VAE in latent diffusion models. Second, on out-of-distribution medical images (*e.g.*, histopathology) (Kumar et al., 2020; Naylor et al., 2019), the natural-image priors of standard diffusion backbones result in misaligned instance boundaries. We provide detailed quantitative results (Tabs. 14 and 15) and qualitative failure cases (Fig. 15) for these scenarios in Appendix E.4.

**Computational Overhead.** In Tab. 7, TRACE introduces minimal additional cost across different evaluation settings. For training-free unsupervised instance segmentation in Tab. 1(a), TRACE refines each image's masks during inference, which increases latency by only about 2%

Table 7: Computational Overhead of TRACE.

| Phase (Dataset) | TRACE (SDXL) | TRACE (SD3.5-L) |
|---|---|---|
| Train (ImageNet) | 8 days | 10 days |
| Test (VOC2012) | 0.1 hrs | 0.2 hrs |
| Test (COCO2014) | 2.0 hrs | 2.4 hrs |
| VRAM Usage | 20 GB | 32 GB |

compared to the ProMerge (Li & Shin, 2024). In contrast, for weakly-supervised panoptic segmentation (Tab. 1(b), Tab. 2), TRACE is used only once during training to refine pseudo instance or panoptic masks before the teacher network (*e.g.*, Mask2Former) is trained, so there is no runtime overhead at inference.

## 7 CONCLUSION

TRACE demonstrates that text-to-image diffusion models naturally encode recoverable instance structure. By locating the Instance Emergence Point, extracting boundaries through self-attention, and compressing them into a fast one-step decoder, TRACE delivers sharp and connected instance edges in real time without any prompts, points, or boxes, or masks. Our extensive evaluation across diverse diffusion architectures confirms that this capability is intrinsic to the generative diffusion prior, consistently yielding superior instance boundary precision and topological connectivity compared to non-diffusion baselines and conventional edge detectors. These edges act as annotation-free instance seeds that boost both interactive systems like SAM and unsupervised/weakly-supervised pipelines, surpassing point- and box-supervised alternatives. Looking forward, the same principle opens opportunities for video panoptic segmentation, medical imaging, and open-vocabulary grouping where text and TRACE can be combined for scalable panoptic perception.

## 8 ACKNOWLEDGMENTS

This work was partly supported by the KHIDI grant funded by the Korean government (MOHW) [No.RS-2025-02307233], the IITP grants funded by the Korean government (MSIT) [No.RS-2025-02305581, No.RS-2025-25442338 (AI Star Fellowship-SNU), and No.RS-2021II211343 (SNU AI)], the Research grant from SNU, and the Strategic Hub grant for International Research Collaboration of SNU.

Kyungsu Kim is affiliated with the School of Transdisciplinary Innovations, Department of Biomedical Science, Interdisciplinary Program in Artificial Intelligence (IPAI), Medical Research Center, and AI Institute at SNU.

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

# Appendix

## A    LLM USAGE DISCLOSURE

We used a large-language model (LLM) only for minor text editing, such as correcting typos and adjusting wording to a formal academic tone. The LLM was not involved in research ideation, experimental design, implementation, or analysis. All scientific content and claims were conceived and written by the authors, who take full responsibility for the entire paper.

## B    COMPREHENSIVE REVIEW OF RELATED WORK

### B.1    DIFFUSION MODELS IN SEGMENTATION

Text-to-image diffusion models (Peebles & Xie, 2023; Bao et al., 2023; Chen et al., 2024; Podell et al., 2023) are trained on image-caption pairs to learn complex visual-text relationships, with recent versions such as SDXL (Podell et al., 2023) and SD3 (Esser et al., 2024) offering finer control through transformer-based architectures. These models generate refined images by iteratively denoising latent representations, making them effective for detailed image synthesis. Recently, diffusion models have shown potential for segmentation tasks by generating pseudo-masks and aiding unsupervised segmentation. Relevant methods include:

- **Diffusion-based Segmentation.** Recent approaches repurpose the internal representations of diffusion models for segmentation tasks. DiffCut (Couairon et al., 2024) and DiffSeg (Tian et al., 2024) apply clustering algorithms, such as normalized cuts or K-means clustering, directly to self-attention maps to generate class-agnostic semantic masks. However, these methods often require heuristic threshold tuning and struggle to distinguish adjacent instances of the same class. EmerDiff (Namekata et al., 2024) explores the emergence of pixel-level semantic knowledge by aggregating attention features across timesteps and finding semantic correspondences. While effective for semantic segmentation, EmerDiff (Namekata et al., 2024) focuses on gathering stable semantic signals rather than detecting the transient structural boundaries between instances. Consequently, these clustering- and aggregation-based methods inherently prioritize semantic grouping (merging same-class pixels) over instance separation. By contrast, TRACE specifically targets the Instance Emergence Point (IEP; Sec. 3.2) and leverages Attention Boundary Divergence (ABDiv; Sec. 3.3) to capture high-frequency boundary signals, enabling the precise delineation of individual instances without annotations.

- **Diffusion-Driven Dataset Generation.** Several approaches leverage diffusion models to synthesize training data with pixel-level annotations. DatasetDM (Wu et al., 2023) and Dataset Diffusion (Nguyen et al., 2023) generate synthetic image-mask pairs to train downstream segmentation networks. MosaicFusion (Xie et al., 2025) adopts a tiling strategy for instance segmentation data: it generates single-object images from simple prompts (*e.g.*, "A photo of a cat") and composites them into a $2 \times 2$ grid, assigning distinct instance IDs based on grid positions. While effective for data augmentation, this mosaic approach artificially avoids the challenge of segmenting naturally adjacent or overlapping objects within a coherent scene. In stark contrast, TRACE is not a data-synthesis pipeline but a *decoding framework*. We reveal that the early denoising steps of a pretrained text-to-image model already contain rich, recoverable instance-level cues. Unlike MosaicFusion's reliance on synthetic composition, TRACE directly extracts precise instance boundaries from a single real image by exploiting the intrinsic Instance Emergence Point (IEP) and Attention Boundary Divergence (ABDiv), successfully separating complex adjacent instances without additional training or prompts.

Existing diffusion-based segmentation methods (Couairon et al., 2024; Tian et al., 2024) focus on later timesteps in the inversion process, when image structures are nearly complete. However, this focus limits their ability to identify and separate multiple instances, as instance-specific information appears in the early timesteps but fades as the model builds the overall semantic structure. This novel use of early diffusion features enables TRACE to capture instance-level information and perform instance segmentation effectively.

## B.2 OPEN-VOCABULARY SEGMENTATION

Recent methods combining multimodal models (MLLM/VLM) with segmentation frameworks, such as SAM (Kirillov et al., 2023) or its unsupervised counterpart UnSAM (Wang et al., 2024b) have enabled open-vocabulary panoptic segmentation. Numerous approaches utilizing CLIP (Radford et al., 2021) demonstrated that free-form text prompts can guide segmentation (Zhou et al., 2022; Cha et al., 2023; Jo et al., 2024b; Barsellotti et al., 2025); however, these methods generally underperform compared to specialized segmentation models that rely on precise mask annotations (Cheng et al., 2022). More recent advancements, such as Grounding DINO (Liu et al., 2024b) and Ferret (You et al., 2023), have incorporated box annotations and language models to improve detection. Despite this progress, these models often struggle to generate accurate instance seeds from multi-tag inputs, failing to distinguish visually similar classes. Extending its efficacy beyond unsupervised and weakly-supervised segmentation (Tabs. 1 and 2), TRACE effectively complements open-vocabulary frameworks by injecting robust instance-aware boundaries, yielding consistent performance improvements across multiple benchmarks (see Appendix E.3 and Tab. 13).

## B.3 INSTANCE SEGMENTATION

Instance segmentation (IS) aims to delineate and label individual object instances within images. Traditional IS approaches rely on pixel-level annotations, which can be resource-intensive, particularly for large datasets. More recent work has expanded into unsupervised methods to address scalability concerns. Meanwhile, panoptic segmentation (PS) methods, designed to handle both semantic and instance segmentation in a unified task, often serve as competitive baselines for instance segmentation capabilities. Panoptic models distinguish between "things" (countable objects) and "stuff" (amorphous regions) and can separate multiple instances of the same class. Here, we categorize recent IS and PS methods based on their reliance on annotations.

**Fully-supervised Segmentation.** Many established methods (Cheng et al., 2022; Li et al., 2022b; Zhang et al., 2021; Cheng et al., 2021; Li et al., 2021; Kirillov et al., 2019; Xu et al., 2023), such as Mask2Former (Cheng et al., 2022) and Panoptic SegFormer (Li et al., 2022b), require dense, pixel-level annotations(*i.e.*, masks) to achieve accurate instance segmentation but face scalability issues due to high annotation costs. These models are designed for panoptic segmentation, but serve as baselines for instance segmentation. We refer to these methods as "panoptic models".

**Point- and Box-Supervised Segmentation.** These weakly-supervised methods (Li et al., 2023b; Fan et al., 2022; Li et al., 2018; 2023a; Jiang et al., 2024) aim to reduce annotation costs by using less detailed - but instance-specific - labels like points or boxes for each instance. These methods (Li et al., 2023b; Fan et al., 2022) typically operate in two stages: first, they generate pseudo-panoptic masks from weak annotations, and then a panoptic model (*e.g.*, Mask2Former) is trained using these pseudo-masks instead of ground-truth annotations. Although annotation effort is reduced, the reliance on manual points and boxes still presents a substantial cost. However, TRACE is annotation free.

**Tag-Supervised Segmentation.** Tag-supervised instance segmentation uses image-level tags (*e.g.*, "person") without any instance-level details (*e.g.*, location of each person). While only a single image-level tag is required per image, depending on the number of instances, multiple instance-level annotations may be required. Hence, tag supervision is significantly cheaper than instance-level annotations.

- **Instance Segmentation with Class Activation Maps (CAM) (Zhou et al., 2016).** CAM-based instance segmentation methods (Kim et al., 2022; Zhou et al., 2018) leverage heatmap and class tags for rough localization of class-related regions. However, CAM was initially designed for semantic segmentation, and its lack of precision in localizing specific instances limits its effectiveness for instance-level segmentation.

- **Weakly-supervised Semantic Segmentation (WSS).** WSS methods (Yang et al., 2025b; Rong et al., 2023; Kim et al., 2023; Kweon et al., 2023; Deng et al., 2023; Yi et al., 2023; Jo et al., 2023; Zhu et al., 2023; Lin et al., 2023; Ru et al., 2023; Xu et al., 2022; Li et al., 2022a; Liu et al., 2022; Chen et al., 2022; Xie et al., 2022; Fan et al., 2018) use image-level tags or captions and offer promising results in class-level segmentation. However, WSS

alone cannot produce instance-level masks without further refinement. TRACE provides the instance-level refinement that can extend WSS methods to tag-supervised PS methods.

We note that the use of class tags is the minimal possible annotation which allows models to focus on specific classes in an image, crucial for evaluating performance in PS tasks. Methods that do not incorporate class tags (*e.g.*, U2Seg (Niu et al., 2024)) are excluded from our comparisons with PS methods, as they cannot produce specific class labels. U2Seg attempts unsupervised panoptic segmentation by combining MaskCut (Wang et al., 2023a) for unsupervised instance segmentation with STEGO (Hamilton et al., 2022) for unsupervised semantic segmentation. However, without tag supervision, U2Seg cannot generate correct class labels, generating labels such as "class 1" instead of "cat 1", which restricts its applicability. Some methods, like JTSM (Shen et al., 2021), use class tags but demonstrate limited PS performance, further indicating a gap in this domain. Remarkably, TRACE+WSS, which uses only tag supervision in the WSS method, outperforms methods using point supervision.

**Unsupervised Instance Segmentation (UIS).** UIS aims to perform instance segmentation without relying on any annotations. Existing UIS methods can be broadly categorized based on their architectural foundation: DINO-based methods (Niu et al., 2024; Wang et al., 2024a; 2023a; Li & Shin, 2024) like MaskCut (Wang et al., 2023a) and ProMerge (Li & Shin, 2024) apply feature clustering with the graph cut algorithm on pretrained self-supervised vision transformer(*i.e.*, DINO (Caron et al., 2021)) backbones to separate instances. SOLO-based approaches (Wang et al., 2022; Ishtiak et al., 2023) rely on CNN-based (*e.g.*, DenseCL (Wang et al., 2021)) pseudo-mask generation but show poor performance and requires training from scratch, making them computationally expensive. UIS serves as a direct baseline for TRACE, as these methods attempt instance separation without any labels. TRACE stands out from existing instance segmentation literature by taking a fully unsupervised, edge-oriented approach instead of clustering features via graph cut. Also, instead of DINO or SOLO-based backbones, TRACE leverages a generative diffusion model.

## B.4 Additional Analysis of Unsupervised Instance Segmentation

Our method tackles the root causes of problems. **Problem A. Adjacent instance merging** stems from UIS methods (Wang et al., 2023a) relying on semantic backbones Caron et al. (2021), lacking instance-level distinction. We resolve this using instance-aware diffusion self-attention maps via IEP/ABDiv. **Problem B. Single instance fragmentation** arises from fixed hyperparameters $(\tau, n)$ in graph-cut methods (Wang et al., 2023a), leading to inconsistent granularity. We resolve these issues via non-parametric instance edge & BGP. Figure 20, Figure 21, and Figure 16 provide qualitative validation. Appendix B.4 and Table 8 give detailed analysis.

| Component | Main Role | Contributions to Problem A | Contributions to Problem B |
|---|---|---|---|
| IEP | Finds optimal step for instance boundaries | Separates adjacent instances in feature space | Provides initial edge information for merging |
| ABDiv | Converts self-attention maps to edge maps | Detects edge seeds between adjacent instances | Provides instance-aware edge maps for merging |
| Distill | Enhances connectivity in edge maps | Completes edges between adjacent instances | Refines instance edges to resolve fragmentation |
| BGP | Instance-aware random walk propagation | Separates instances using refined edges | Propagates and merges fragmented instance masks |

Table 8: **Component contributions to solve (A) adjacent instance merging and (B) single instance fragmentation.** Each step builds on the previous one, with BGP finalizing adjacent instance separation and fragmentation resolution using the refined edges generated by IEP, ABDiv, and fine-tuning.

### B.4.1 Problem A. Adjacent Instance Merging

**Root Cause of Problem A: Limitations of Existing Backbones.** Unsupervised Instance Segmentation (UIS) methods (Niu et al., 2024; Wang et al., 2024a; 2023a; Li & Shin, 2024) depend on pretrained "backbones" (*e.g.*, DINOv1 (Caron et al., 2021)) to extract feature maps, which are subsequently partitioned into instance masks through techniques like graph cut (Wang et al., 2023b). However, these backbones were originally designed to distill information primarily about *semantic* segmentation—identifying and classifying foreground objects from the background, rather than distinguishing individual instances within a class. This inherent design limitation makes it difficult to achieve true instance separation (e.g., distinguishing two adjacent boats in the top of Fig. 12) using only semantic features.

**MLLM Backbones Share This Limitation, and More Parameters Do Not Resolve It.** When prompted with "Describe this image" in Fig. 8, LLaVA-13B's self-attention maps only capture semantic features, while diffusion models (*e.g.*, SD1.5) separate instances, a capability absent in other foundation models. In Tab. 5, diffusion backbones substantially outperform CLIP/DINO/MLLM counterparts with comparable and fewer parameters.

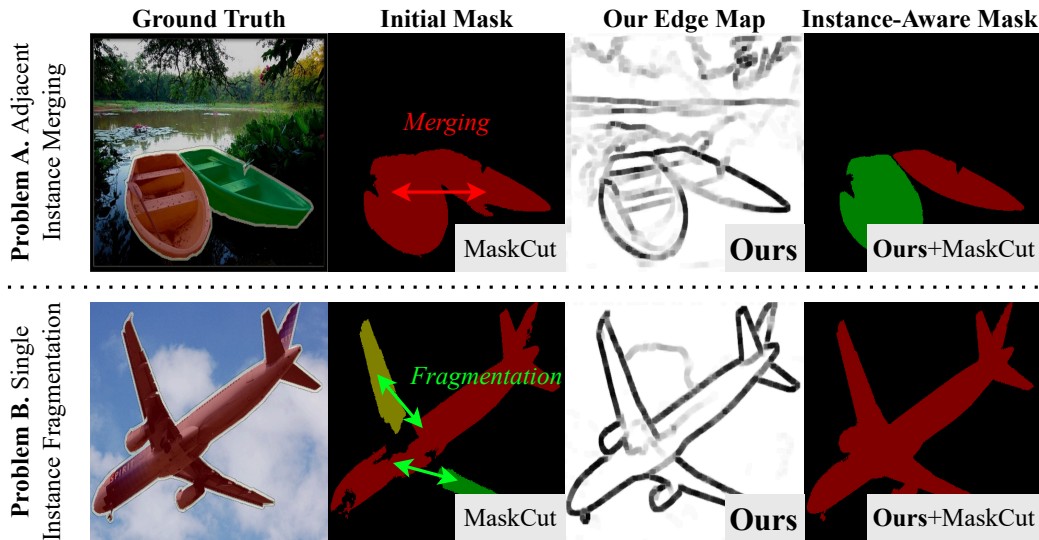

Figure 12: **Additional examples of problem A and B and our solution.**

**Our Solution for Problem A.** To overcome the limitations of semantic-oriented backbones, we introduce TRACE, a framework that leverages the inherent spatial attention in diffusion models to generate instance-aware edges without requiring pixel-level annotations. TRACE harnesses self-attention maps from pretrained diffusion models, which capture instance boundaries through attention mechanisms tuned during pretraining. By introducing two novel metrics, IEP and AB-Div, TRACE autonomously identifies and reinforces instance boundaries, ensuring that adjacent instances are effectively separated, even when traditional feature maps fall short. Unlike conventional UIS methods (Niu et al., 2024; Wang et al., 2024a; 2023a; Li & Shin, 2024) that rely on semantic segmentation backbones, TRACE uses a self-supervised fine-tuning step that enhances edge connectivity, achieving precise instance delineation across classes. This enables TRACE to address the issue of adjacent object merging by focusing specifically on instance edges, demonstrating superior separation accuracy across challenging datasets (*e.g.*, VOC (Everingham et al., 2010), COCO (Lin et al., 2014)), as shown in our experimental results (see Tab. 1 and Tab. 2). Our diffusion-based approach addresses the adjacency issue by focusing on instance-level edge generation rather than relying solely on semantic features, making it a scalable solution for UIS.

### B.4.2 PROBLEM B. SINGLE INSTANCE FRAGMENTATION

**Root Cause of Problem B: Limitations of Graph Cut.** Graph-cut based UIS methods (Wang et al., 2023b;a; Li & Shin, 2024) require a hyperparameter, $\tau$, which acts as a threshold for the initial graph construction. In this approach, the image is represented as a graph, where each pixel (or patch) corresponds to a vertex, and each edge represents the degree of similarity (*i.e.*, affinity) between two pixels (or patches). The affinity $A_{i,j}$ between vertices $i$ and $j$ is measured using the cosine similarity $\mathcal{S}_{i,j}$ of their respective feature maps. If $\mathcal{S}_{i,j} \geq \tau$, vertices $i$ and $j$ are connected by an edge; otherwise, they are not connected.

Intuitively, $\tau$ determines the sensitivity of the UIS method to similarities between feature maps. For instance, if $\tau$ is close to 1, only pixels with nearly identical feature maps are considered connected. Conversely, if $\tau = 0$, every pixel is connected to every other pixel. The graph-cut algorithm then partitions the graph into two partitions—a foreground partition and a background partition—by minimizing the number of edges that need to be removed. The foreground object is identified using a heuristic rule, and the corresponding partition is presented as an instance mask. For the UIS

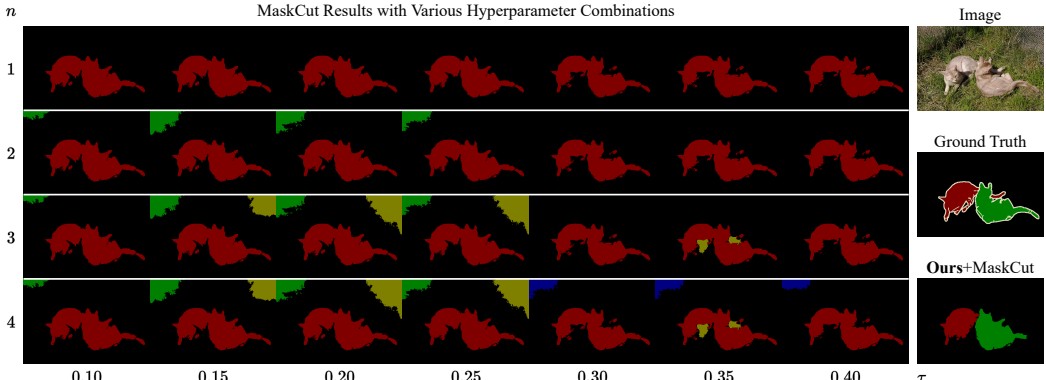

Figure 13: **Drawbacks of MaskCut with various** $n, \tau$ **combinations.** Due to the semantic-oriented nature of the DINO backbone used in MaskCut, increasing $n$ or decreasing $\tau$ does not result in the successful separation of the two adjacent instances. Rather, changing the hyperparameters leads to fragmentation and detection of false positives in the background.

method to detect at most $n$ distinct instances, this process needs to be repeated for $n$ iterations, where $n$ is a fixed hyperparameter.

The value of $\tau$ impacts the granularity of the instance masks produced by graph-cut based methods (see Fig. 13). A fixed $\tau$ often leads to fragmentation of single instances or merging of multiple instances, as it cannot be dynamically adjusted for each image. Additionally, methods that fix iteration counts, $n$, cannot detect more than $n$ instances, failing in instance-dense images. This fixed tuning for $\tau$ and $n$ may work on the average case, but does not generalize well across different images.

**Our Solution for Problem B.** By contrast, TRACE predicts edges between adjacent instances without tuning hyperparameters (*e.g.*, $n, \tau$) related to instance count, making it more flexible and robust. Experimentally, TRACE-generated edges effectively resolve fragmentation and separate adjacent instances, achieving both conceptual and practical advantages. Since TRACE-generated edges accurately delineate each instance, we can use the edge map to construct a transition probability matrix for a random walk on the pixel space. This spreads out the masks within the edges (increased IoU between masks of the same instance) while effectively stopping masks from spreading over the edges (restricted IoU between masks of distinct instances, even if the instances are adjacent). As a result, our **edge-generation approach** allows TRACE to handle both adjacent instance merging and single instance fragmentation, enhancing existing UIS methods (Li & Shin, 2024; Wang et al., 2023a).

## C    METHOD DETAILS

### C.1    REPRODUCIBILITY

Training follows standard diffusion model fine-tuning regardless of the backbone. During the one-step distillation stage (Sec. 3.4), LoRA adapters with rank 64 are optimized using Adam with a weight decay of 4e-5. Input images are randomly resized to the native resolution of each backbone, namely $512 \times 512$ for SD1.5 and $1024 \times 1024$ for SDXL and SD3.5-L. For UIS and WPS, dataset splits, augmentations, and evaluation protocols strictly follow the corresponding prior works (Jo et al., 2024a; Li et al., 2023b; Wang et al., 2023a) to ensure a fair comparison; hyperparameters not specified above are inherited from the respective baselines.

### C.2    DIFFUSION MODELS AND FLOW MATCHING MODELS

Diffusion models (Rombach et al., 2022; Podell et al., 2023) and Flow Matching models (Esser et al., 2024; Lipman et al., 2022) are generative models that learn to produce realistic data by reversing a gradual noising process. We use the term *diffusion model* to refer to both, as our method is compatible with either. Starting from a clean image $X_0 \sim p_0$, noise is progressively added over time to obtain $X_t$:

$$X_t = \alpha_t X_0 + \sigma_t \epsilon, \quad \epsilon \sim \mathcal{N}(0, I) \tag{3}$$

Here, $\alpha_t$ and $\sigma_t$ vary with time $t$ and control how much of the image and noise are mixed. Early steps keep the image mostly intact; later steps produce nearly pure noise. This design enables the model to learn different levels of structure across time, from coarse to fine.

A neural network $\epsilon_\theta(X_t, t)$ is trained to predict the noise $\epsilon$, minimizing the loss:

$$\theta^\star = \arg\min_\theta \mathbb{E}_{X_0, t, \epsilon} \|\epsilon_\theta(X_t, t) - \epsilon\|_2^2 \tag{4}$$

After training, the original image can be approximately recovered by removing the predicted noise:

$$\hat{X}_0 = \frac{1}{\alpha_t} \left( X_t - \sigma_t \epsilon_{\theta^*}(X_t, t) \right) \tag{5}$$

We denote this forward-and-reverse process as $\texttt{Reconstruct}(\epsilon_{\theta^*}, X_0, t)$.

Flow Matching models follow a similar principle but learn the velocity $\dot{X}_t$ of the diffusion process using a network $v_\theta$. They minimize $\mathbb{E}_{X_0, t, \epsilon} \|v_\theta(X_t, t) - (\dot{\alpha}_t X_0 + \dot{\sigma}_t \epsilon)\|_2^2$ with reconstruction given by $\hat{X}_0 = \frac{1}{\dot{\alpha}_t} \left( v_\theta(X_t, t) - \dot{\sigma}_t \epsilon \right)$.

## C.3 SELF-ATTENTION ACCUMULATION

For a noised image $X_t$ at time step $t$, the self-attention mechanism computes spatial dependencies using weights $W_Q, W_K$:

$$Q_t = X_t W_Q, \ K_t = X_t W_K \in \mathbb{R}^{HW \times d} \tag{6}$$

$$SA(X_t) = \text{softmax}\left( \frac{Q_t K_t^T}{\sqrt{d}} \right) \in [0, 1]^{HW \times HW} \tag{7}$$

where $d$ is the dimensionality of the attention heads, and $H, W$ are the height and width of the self-attention map.

In text-to-image diffusion models, multiple self-attention maps $SA_1, \ldots, SA_N$ are produced at various resolutions $w_1, \ldots, w_N$ ($w_i$ is the width of the $i$th SA map). We accumulate these maps by upsampling each to the resolution of the highest-resolution map, followed by averaging across all maps:

$$SA[i, j, :, :] = \frac{1}{N} \sum_{k=1}^{N} SA_k \left[ \frac{i}{\delta_k}, \frac{j}{\delta_k}, :, : \right] \tag{8}$$

$$\delta_k = \frac{\max\limits_{1 \le j \le N} w_j}{w_k} \tag{9}$$

The hook function $\mathcal{H}(\epsilon_\theta, X_0, t)$ extracts all self-attention maps evaluated in the forward pass at timestep $t$ given input image $X_0$ and returns the accumulated attention map in Eq. 8.

## C.4 DETAILS OF IEP AND ABDIV

**Algorithm 1 (IEP; Sec. 3.2):** Identifies the optimal diffusion timestep $t^*$ for instance separation by locating the peak in the Kullback-Leibler (KL) divergence between consecutive attention maps. As derived in Appendix D, this peak theoretically corresponds to the point of maximum Fisher information with respect to the noise level, marking the rapid emergence of structural instance cues. Empirical consistency of this metric across diverse models and random seeds is further validated in Appendix E.2.

---

**Algorithm 1** IEP : $X_0$ (Image) $\mapsto$ SA$_{\text{inst}}$

---

**Require:** Image $X_0$, Sequence of timesteps $\tau_0 < \tau_1 < \cdots < \tau_N$, Frozen Diffusion Model $\epsilon_{\theta^\star}(\cdot, \cdot)$
 1: Initialize: KL-gap$\leftarrow$ 0, SA$_{\text{inst}} \leftarrow \emptyset$
 2: $\hat{X}_0 \leftarrow$ Reconstruct$(\epsilon_{\theta^\star}, X_0, \tau_1)$                                                Hook $\mathcal{H}$
 3: $SA(X_{\tau_0}) \leftarrow \mathcal{H}(\epsilon_{\theta^\star}, X_0, \tau_0)$
 4: **for** $n = 1$ to $N$ **do**
 5:     $\hat{X}_0 \leftarrow$ Reconstruct$(\epsilon_{\theta^\star}, X_0, \tau_n)$                                     Hook $\mathcal{H}$
 6:     $SA(X_{\tau_n}) \leftarrow \mathcal{H}(\epsilon_{\theta^\star}, X_0, \tau_n)$
 7:     **if** KL-gap$< D_{\text{KL}}(SA(X_{\tau_{n-1}}) \parallel SA(X_{\tau_n}))$ **then**
 8:         KL-gap$\leftarrow D_{\text{KL}}(SA(X_{\tau_{n-1}}) \parallel SA(X_{\tau_n}))$
 9:         SA$_{\text{inst}} \leftarrow$ normalized $SA(X_{\tau_n})$
10:     **end if**
11: **end for**
12: return SA$_{\text{inst}}$

---

**Algorithm 2 (ABDiv; Sec. 3.3):** Transforms instance-aware self-attention maps into pseudo-edge maps without annotations. Crucially, this algorithm implements a reliability-based filtering step: pixels with ABDiv scores in the intermediate range $(\mu - \sigma, \mu + \sigma)$ are labeled as uncertain ($E_{i,j} = -1$) because they typically correspond to ambiguous texture gradients or noise rather than definitive instance boundaries. By explicitly masking these regions, we prevent the introduction of label noise during the subsequent self-distillation phase, ensuring that the edge decoder learns only from high-confidence boundary signals.

---

**Algorithm 2** ABDiv: $\mathrm{SA_{inst}} \mapsto E$ (Pseudo Instance Edge)

---

**Require:** Self-Attention Map $\mathrm{SA_{inst}}$
1: Initialize: $E_0 \leftarrow \mathbf{0}_{H \times W}$
2: **for** $(i, j)$ in $[H] \times [W]$ **do**
3:    $E_0[i, j] \leftarrow \mathrm{ABDiv}(i, j; \mathrm{SA_{inst}})$
4: **end for**
5: $\mu \leftarrow$ mean of $E_0$
6: $\sigma \leftarrow$ standard deviation of $E_0$
7: $E = \begin{cases} \mathbf{1} \cdot [E_0 > \mu + \sigma] & \text{Edge Pixel} \\ \mathbf{0} \cdot [E_0 < \mu - \sigma] & \text{Interior Pixel} \\ (-\mathbf{1}) \cdot [\text{otherwise}] & \text{Uncertain Pixel} \end{cases}$
8: **return** $E$

---

## C.5    Details of One-step Self-Distillation with Edge Decoder

Algorithm 3 (Sec. 3.4): Fine-tunes the transformer layers of text-to-image diffusion models and trains a new edge generator $\mathcal{G}_\phi$ from scratch. For the architecture of $\mathcal{G}_\phi$, we use a lightweight CNN decoder for simplicity, and in Tab. 9 we show that replacing it with heavier designs such as U-Net or Mask2Former-style decoders yields negligible gains, indicating that most of the instance-boundary detail already resides in the TRACE edge cues themselves.

---

**Algorithm 3** One-step Self-Distillation of TRACE

---

**Require:** Image Dataset $\mathcal{I}$, Pseudo-Edge Set $\{E_I\}_{I \in \mathcal{I}}$
         Edge Generator $\mathcal{G}_\phi$
         Pretrained Text-to-Image Diffusion Model $\epsilon_\theta$
1: **for** Epoch in $1 \dots N$ **do**
2:   $\mathcal{L} \leftarrow 0$
3:   **for** $I$ in $\mathcal{I}$ **do**
4:     $\hat{I} \leftarrow \mathrm{Reconstruct}(\epsilon_\theta, I, \tau = 0)$                       Hook $\mathcal{H}$
5:     $\hat{E} \leftarrow \mathcal{G}_\phi(\mathcal{H}(\epsilon_\theta, I, \tau = 0))$            Predicted Edge
6:     $E \leftarrow E_I$                                   Pseudo-Edge
7:     $\mathcal{L} \leftarrow \mathcal{L} + \|I - \hat{I}\|^2 + \mathrm{DiceLoss}(E, \hat{E})$
8:   **end for**
9:   Backprop on $\mathcal{L}(\theta, \phi)$
10: **end for**

---

Dice loss is computed as follows:

$$\mathrm{DiceLoss}(E, \hat{E}) = 1 - \frac{2 \sum W_{i,j} E_{i,j} \hat{E}_{i,j}}{\sum W_{i,j} E_{i,j}^2 + \sum W_{i,j} \hat{E}_{i,j}^2} \tag{10}$$

and the weighting $W_{i,j} = \mathbb{1}[E_{i,j} \neq -1]$ excludes uncertain pixels. We exclude uncertain pixels (where $E = -1$) from the dice loss computation via $W$ to allow the model to focus on confident edge and interior points.

## C.6 DETAILS OF BOUNDARY-GUIDED PROPAGATION

---

**Algorithm 4** UIS/WSS Segmentation Masks with TRACE

---

**Require:** Image $I$
        Trained Edge Decoder $\mathcal{G}_{\phi^*}$
        Fine-Tuned Diffusion Model $\epsilon_{\theta^*}$
        Masks $M_1, \ldots, M_N$ from a UIS/WSS Method
1: $\hat{I} \leftarrow$ `Reconstruct`$(\epsilon_{\theta^*}, I, t = 0)$                                   Hook $\mathcal{H}$
2: $\hat{E} \leftarrow \mathcal{G}_{\phi^*}(\mathcal{H}(\epsilon_{\theta^*}, I, t = 0))$                        Predicted Edge
3: Identify Connected Components using $\hat{E}$ and CCL
4: $M_n^* \leftarrow$ Propagate Mask $M_n$ via BGP                           Figure 6
5: Merge Masks with an IoU $> \tau_{\text{BGP}}$
6: Return Final Masks

---

We apply the random-walk methods proposed in (Ahn & Kwak, 2018; Ahn et al., 2019). Here, we outline a high-level overview of the propagation technique.

1. **Sparse Affinity Construction:** From a pseudo-edge map $\hat{E}$, construct sparse affinity matrix $A_{\text{sparse}}$ by calculating affinities between each pixel and its neighbors along predefined paths. Paths with edges (indicating boundaries) have low affinities, while edge-free paths have affinities closer to 1.

2. **Sparse to Dense Affinity:** Convert $A_{\text{sparse}}$ into a dense matrix $A_{\text{dense}}$ by filling in affinities for every pixel pair. For symmetric consistency, if a path exists between pixels $i$ and $j$, then both $A_{\text{dense}}[i, j]$ and $A_{\text{dense}}[j, i]$ are assigned the same value.

3. **Seed Propagation:** Compute the transition matrix $T = D^{-1} A_{\text{dense}}^{\circ\beta}$, where $D$ normalizes rows of $A_{\text{dense}}^{\circ\beta}$. After $t$ iterations, updated masks are generated via:

$$\text{vec}(M_c^*) = T^t \cdot \text{vec}(M_c \odot (1 - \hat{E})), \tag{11}$$

where $(1 - \hat{E})$ prevents edge pixels from influencing neighbors, focusing propagation on non-edge regions.

This method creates more coherent and complete instance masks.

## D INFORMATION THEORETIC VIEW OF IEP

**Setup and Assumptions.** Fix a query pixel $i$ and let the $i$-th self-attention row at step $t$ be the softmax distribution:

$$p_t(j \mid i) = \frac{\exp(\gamma_t s_{ij})}{Z_t(i)}, \quad Z_t(i) = \sum_j \exp(\gamma_t s_{ij}) \tag{12}$$

where $s_{ij} = \frac{1}{\sqrt{d}} \mathbf{q}_i^\intercal \mathbf{k}_j$ is a *time-invariant* "clean" similarity of $i, j$ (similarity in the clean image) and $\gamma_t > 0$ is an effective inverse temperature. We assume throughout:

**A.1.** (*Monotone schedule.*) $\gamma_t$ is $C^1$, takes values in $[0, 1]$, and is strictly decreasing along the forward trajectory $t \uparrow$ (noise increases), with $\gamma_{t=0} = 1$ (clean image) and $\gamma_{t=1000} \approx 0$ (pure noise).

**A.2.** (*Bounded logits.*) There exists $S < \infty$ such that $|s_{ij}| \leq S$ for all $i, j$

**A.3.** (*Non-degenerate row.*) For a given $i$, the set of similarity values $\{s_{ij}\}_j$ contains at least wo distinct values.

**High-level Intuition.** Under the standard forward diffusion parameterization $x_t = \alpha_t x_0 + \sigma_t \varepsilon$, $\varepsilon \sim \mathcal{N}(0, I)$, the signal-to-noise ratio (SNR) $\alpha_t^2 / \sigma_t^2$ decreases along the forward trajectory as $t \uparrow$. Thus, it is natural and empirically accurate to model $\gamma_t$ as a strictly decreasing function of $t$ such that

$\gamma_t \propto \mathrm{SNR}(t)$. At pure noise (*e.g.*, $t = 1000$), a lower value of $\gamma_t$ gives higher entropy; at pure signal (*e.g.*, $t = 0$), a high value of $\gamma_t$ yields peaked assignments.

These minimal assumptions suffice for the identities below.

## D.1 ROW ENTROPY AND ITS TIME DERIVATIVE

For the row entropy $H_t(i) = -\sum_j p_t(j \mid i) \log p_t(j \mid i)$, classical exponential-family algebra Amari & Nagaoka (2000) gives:

$$H_t(i) = \log Z_t(i) - \gamma_t \mathbb{E}_{p_t(\cdot|i)}[s_{i\cdot}], \quad \frac{\partial H_t(i)}{\partial \gamma_t} = -\gamma_t \operatorname{Var}_{p_t(\cdot|i)}[s_{i\cdot}] \tag{13}$$

because $\frac{d}{d\gamma_t} \mathbb{E}_{p_t(\cdot|i)}[s_{i\cdot}] = \operatorname{Var}_{p_t(\cdot|i)}[s_{i\cdot}]$ and $\frac{d}{d\gamma_t} \log Z_t = \mathbb{E}_{p_t(\cdot|i)}[s_{i\cdot}]$. By the chain rule,

$$\frac{dH_t(i)}{dt} = \frac{dH_t(i)}{d\gamma_t} \dot{\gamma}_t = -\dot{\gamma}_t \gamma_t \operatorname{Var}_{p_t(\cdot|i)}[s_{i\cdot}] \tag{14}$$

Since $\dot{\gamma}_t < 0$ along the forward trajectory (**A.1.**), Eq. 14 says row entropy *increases* from the clean end toward the noise end ("slow-fast-slow" in magnitude, as we soon show), with the rate controlled by the variance of $s_{i\cdot}$ under the current row distribution.

**Endpoint Behavior and Boundedness.** By **A.2.**, the variance of $s_{i\cdot}$ under $p_t(\cdot \mid i)$ is bounded by:

$$\operatorname{Var}_{p_t(\cdot|i)}[s_{i\cdot}] = \sum_j p_t(j \mid i)(s_{ij} - \mathbb{E}_{p_t(j|i)}[s_{ij}])^2 \le S^2 < \infty \tag{15}$$

Hence, $\left| \frac{dH_t(i)}{dt} \right| \le \gamma_t |\dot{\gamma}_t| S^2$.

- **At pure noise.** At the noise end, $\gamma_t \to 0$, so $|dH_t(i)/dt| \to 0$ regardless of the exact variance. This theoretical result matches the small temporal change of near-uniform attention observed before IEP.
- **At pure signal.** At the *clean image* end, a different mechanism makes $|dH_t(i)/dt|$ small: if for $j^* = \arg\max_j s_{ij}$ and the top-2 similarity gap $\Delta_i := s_{ij^*} - \max_{j \ne j^*} s_{ij} > 0$, then $p_t(\cdot \mid i)$ concentrates on $j^*$ and $\operatorname{Var}_{p_t(\cdot|i)}[s_{i\cdot}]$ shrinks.

Empirically, we see small temporal change at both ends and a single interior region of rapid change which aligns with our theory.

## D.2 INTER-STEP KL AND FISHER INFORMATION

**Inter-step KL Peaks at Maximal Fisher information.** Recall that the Fisher information in $\gamma_t$ is defined by

$$\mathcal{I}_i(\gamma_t) := \mathbb{E}\left[ \left( \frac{\partial}{\partial \gamma_t} \log p_t \right)^2 \mid \gamma_t \right] = \mathbb{E}\left[ \left( \frac{\partial}{\partial \gamma_t} (\gamma_t s_{ij} - \log Z_t) \right)^2 \mid \gamma_t \right] = \underbrace{\mathbb{E}\left[ \left( s_{ij} - \mathbb{E}_{p_t(\cdot|i)}[s_{i\cdot}] \right)^2 \mid \gamma_t \right]}_{=\operatorname{Var}_{p_t}[s]}$$

As in our implementation, we consider the KL divergence between consecutive attention rows at steps $t - \Delta t$ and $t$. For small $\Delta t$, a second-order expansion of the row KL divergence yields:

$$\mathrm{KL}(p_{t-\Delta t}(\cdot \mid i) \| p_t(\cdot \mid i)) = \frac{1}{2} \underbrace{(\gamma_t - \gamma_{t-\Delta t})^2}_{\Delta \gamma_t} \underbrace{\operatorname{Var}_{p_t}[s]}_{=\mathcal{I}_i(\gamma_t)} + o\left((\Delta \gamma_t)^2\right) \tag{16}$$

where $\mathcal{I}_i(\gamma_t)$ is the Fisher information of the one-parameter family $p_t(j \mid i) \propto \exp(\gamma_t s_{ij})$ with respect to $\gamma_i$.

Thus, the Instance Emergence Point $t^*$ that maximizes the row-wise temporal KL coincides (to second order) with the *point of maximal Fisher information* for that row. Averaging rows (as we do operationally when measuring the KL over full maps) preserves the same interpretation at the

map level. We also remark that *any* $f$-divergence has the same local quadratic form with the Fisher information Nielsen & Hadjeres (2019) by a difference of a factor $c_f > 0$,

$$D_f(p_{t-\Delta t}(\cdot \mid i)\|p_t(\cdot \mid i)) = c_f \cdot (\Delta\gamma_t)^2 \underbrace{\mathrm{Var}_{p_t}[s]}_{=\mathcal{I}_i(\gamma_t)} + o\big((\Delta\gamma_t)^2\big),$$

explains why KL, JSD, and other $f$-divergences all produce the same IEP location (up to small higher-order effects), as highlighted in Tab. 4.

# E COMPREHENSIVE ANALYSIS AND DISCUSSION

## E.1 ABLATION STUDIES ON DESIGN CHOICES

**Sensitivity to Instance-aware Text Prompts.** All main results of TRACE are obtained without instance-aware prompts (*e.g.*, A photo of two cats), using only a null-text prompt. To assess the potential benefit of such supervision, we conduct an additional experiment in which, during training, we exploit the instance annotations in ImageNet (Krizhevsky et al., 2012) to construct descriptive prompts of the form "A photo of [number of boxes] [class]", and at inference we mirror this setup on COCO (Lin et al., 2014) by building prompts from its instance annotations in the same way while keeping all other components of TRACE fixed. In this setting, $AP^{mk}$ increases slightly from 8.2 (null-text) to 8.3 (instance-count prompt), indicating that explicit instance-aware text information can provide a small but marginal gain compared to the overall improvement brought by TRACE itself. Moreover, under the same conditions as Fig. 8(b), the diffusion trajectories with null-text and instance-count prompts are almost indistinguishable, and we do not observe any systematic shift in the optimal timestep $t^*$. These observations support our claim that TRACE does not rely on instance-aware text supervision: the crucial instance-boundary cues are already encoded in the early diffusion timesteps conditioned on the image alone, with instance-count prompts offering only minor additional refinement.

**Influence of Decoder Capacity.** A natural question is whether sharper instance boundaries actually require a heavier edge decoder. To isolate the effect of decoder capacity, we replace our lightweight edge decoder $\mathcal{G}_\phi$ with two representative alternatives (Ronneberger et al., 2015; Cheng et al., 2022), while keeping the TRACE instance-edge cues (*i.e.*, the accumulated multi-scale self-attention maps $\mathcal{H}(\cdot)$) fixed. As shown in Tab. 9, upgrading the edge decoder from our 0.1 MB 1-layer CNN to a

Table 9: **Effect of edge decoder capacity.** We replace our lightweight CNN edge decoder in TRACE with heavier alternatives and evaluate performance of unsupervised instance segmentation on COCO 2014.

| Method | Params. of Dec. | $AP^{mk}$ |
|---|---|---|
| ProMerge | - | 3.1 |
| + TRACE (U-Net) | 34 MB | 8.2 |
| + TRACE (Mask2Former) | 258 MB | 8.4 |
| + TRACE | 0.1 MB | 8.2 |

34 MB U-Net (Ronneberger et al., 2015) leaves $AP^{mk}$ unchanged (8.2 vs. 8.2), and even a 258 MB Mask2Former-style decoder (Cheng et al., 2022) yields only a marginal gain (8.4 $AP^{mk}$). By contrast, adding TRACE on top of ProMerge (Li & Shin, 2024) already increases $AP^{mk}$ from 3.1 to 8.2. These results indicate that sharp, instance-aware boundaries primarily come from the TRACE instance-edge cues themselves; once these cues are available, a minimal 1-layer CNN decoder is sufficient, and substantially larger decoders bring negligible additional benefit.

**Impact of Uncertainty Masking.** In Algorithm 2 and Eq. 10, pixels whose ABDiv score falls into the "uncertain" range (assigned value $-1$) are ignored from the loss by a binary mask, so that only pixels labeled as 0 (non-edge/background) or 1 (edge/foreground) contribute to the supervision. This strategy follows common practice in weakly-supervised segmentation (Ahn et al., 2019; Jo et al., 2024a), where ambiguous regions are ignored to reduce label noise in pseudo semantic masks. We adopt the same idea for the pseudo instance-edge map $E$, using ABDiv as a reliability cue.

Table 10 quantifies the impact of this design. When we threshold ABDiv only at $\mu$ and treat all pixels as confident (*i.e.*, label $E_{ij} = 1$ if $\mathrm{ABDiv}_{ij} \geq \mu$ and $E_{ij} = 0$ otherwise), the instance segmentation metrics improve over ProMerge (Li & Shin, 2024) but remain limited, and the edge precision at ODS drops to 0.572. This large decrease compared to the proposed $\mu \pm \sigma$ scheme (0.852) indicates a substantial increase in false-positive edge pixels. In contrast, masking uncertain pixels between $\mu - \sigma$ and $\mu + \sigma$ during training not only yields larger gains in $AP^{mk}$ and $AR_{100}^{mk}$, but also improves edge precision by about $1.5\times$ while keeping ODS-recall within $\sim 1\%$ of the $\mu$-only

Table 10: **Ablation on pseudo-labeling schemes for ABDiv (Sec. 3.3).** ODS-Precision and ODS-Recall denote the precision and recall at the optimal dataset scale (ODS) for instance edges, from which the ODS (F-measure) is computed.

| Method | Pseudo-Labeling for ABDiv | $AP^{mk}$ | $AR^{mk}_{100}$ | ODS-Prec. | ODS-Rec. | ODS |
|---|---|---|---|---|---|---|
| ProMerge (Li & Shin, 2024) | – | 3.1 | 7.6 | – | – | – |
| + TRACE | $\mu$ | 6.4 | 11.4 | 0.572 | **0.963** | 0.717 |
| + TRACE | $\mu \pm \sigma$ | **8.2** | **13.1** | **0.852** | 0.950 | **0.889** |

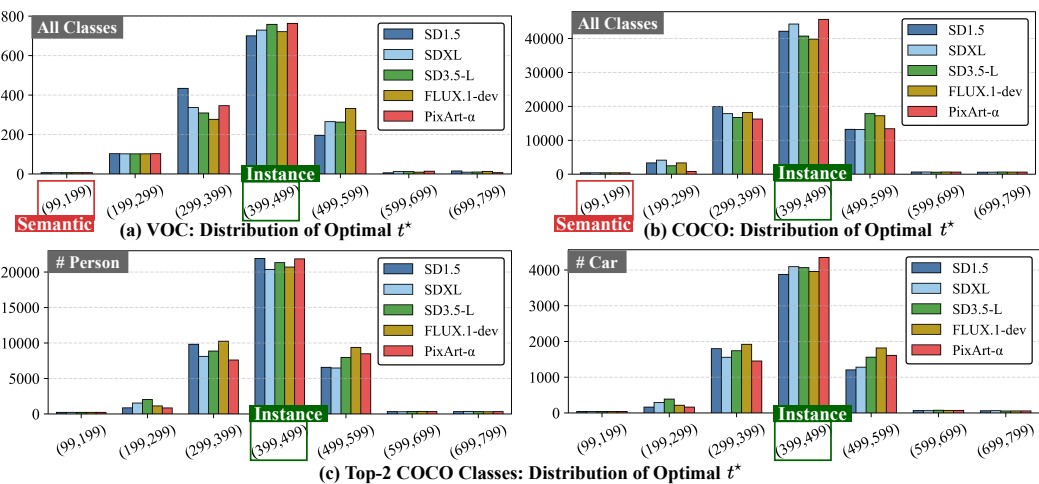

Figure 14: **Histograms of the optimal timestep $t^\star$ across datasets, diffusion backbones, and object categories.** (a) VOC 2012 train set. (b) COCO 2014 train set. (c) Top-2 COCO classes (*person* and *car*). In all cases, the optimal timesteps $t^\star$ for five diffusion models concentrate in a shared instance-aware region, indicating a consistent semantic-to-instance transition and stable IEP behavior across models, datasets, and categories.

variant. This suggests that excluding uncertain regions effectively suppresses noisy edges without sacrificing recall, leading to a more favorable balance between false positives and false negatives in the distilled edge supervision.

### E.2 IN-DEPTH CHARACTERIZATION OF IEP

**Evaluation of Similarity Metrics.** To assess whether our KL criterion is a special case or a broadly effective choice, we extend the comparison in Tab. 4 and evaluate additional similarity measures used as IEP scores, summarized in Tab. 11. Besides symmetric KL and standard regression losses (MSE/L2, MAE/L1), we include an entropy-based score and the Wasserstein-1 distance $W_1$. For the entropy-based metric, we use the absolute entropy difference between two probability maps $p$ and $q$ (*i.e.*, $|H(p) - H(q)|$ with $H(p) = -\sum_i p_i \log p_i$). For the Wasserstein distance, we measure a 1D Wasserstein-1 distance on the flattened distribution. Empirically, KL achieves the highest instance segmentation performance ($AP^{mk} = 9.4$) with moderate latency (3,082 ms per image). Jensen–Shannon divergence (JSD) attains the same $AP^{mk}$ but is substantially slower (5,120 ms), offering no practical advantage over KL. Entropy difference and Wasserstein distance are clearly inferior in $AP^{mk}$ (5.1 for entropy, 6.2 for $W_1$) despite comparable or higher computational cost. We note that JSD can be written as $\mathrm{JSD}(p,q) = \frac{1}{2}\mathrm{KL}(p\|m) + \frac{1}{2}\mathrm{KL}(q\|m)$ with $m = \frac{1}{2}(p+q)$, and is equal to the mutual information between samples and a binary index variable, so this ablation also serves as a mutual-information–style variant of our IEP score. Overall, these results support our choice of KL as the primary IEP metric: it provides the best trade-off between accuracy and effi-

Table 11: Similarity metrics for IEP.

| Metric | Latency/img | VOC | COCO |
|---|---|---|---|
| KL (Ours) | 3,082 ms | **9.4** | **8.2** |
| JSD | 5,120 ms | 9.4 | 8.1 |
| MSE (L2) | 1,232 ms | 3.8 | 3.5 |
| MAE (L1) | 924 ms | 3.5 | 3.3 |
| Entropy | 2,070 ms | 5.1 | 4.1 |
| $W_1$ | 3,434 ms | 6.2 | 5.0 |

Table 12: **Performance of unsupervised instance segmentation on multiple benchmarks.** To ensure a fair comparison with existing UIS models (Wang et al., 2023b;a) fine-tuned on the ImageNet train set (Krizhevsky et al., 2012), we also fine-tune our diffusion model (SD3.5-L (Esser et al., 2024)) and edge generator on the ImageNet train set (Krizhevsky et al., 2012). This allows us to generate TRACE-based instance edges from the ImageNet-trained model and apply them directly to other seven datasets (Everingham et al., 2010; Lin et al., 2014; Gupta et al., 2019; Geiger et al., 2012; Shao et al., 2019; Kirillov et al., 2023) without further fine-tuning.

| Method | VOC 2012 | | COCO 2014 | | COCO 2017 | | LVIS | | KITTI | | Objects365 | | SA-1B | | Average | |
|---|---|---|---|---|---|---|---|---|---|---|---|---|---|---|---|---|
| | $AP^{mk}$ | $AR^{mk}_{100}$ | $AP^{mk}$ | $AR^{mk}_{100}$ | $AP^{mk}$ | $AR^{mk}_{100}$ | $AP^{mk}$ | $AR^{mk}_{100}$ | $AP^{mk}$ | $AR^{mk}_{100}$ | $AP^{mk}$ | $AR^{mk}_{100}$ | $AP^{mk}$ | $AR^{mk}_{100}$ | $AP^{mk}$ | $AR^{mk}_{100}$ |
| TokenCut | 6.1* | 10.6* | 2.7 | 4.6 | 2.0 | 4.4 | 0.9 | 1.8 | 0.3 | 1.5 | 1.1 | 2.1 | 1.0 | 0.3 | 2.0 | 3.6 |
| MaskCut | 5.8* | 14.0* | 3.0* | 6.7* | 2.3* | 6.5* | 0.9* | 2.6* | 0.2* | 1.9* | 1.7* | 4.0* | 0.8* | 0.6* | 2.1 | 5.2 |
| ProMerge | 5.0* | 13.9* | 3.1* | 7.6* | 2.5* | 7.5* | 1.1* | 3.4* | 0.2* | 1.6* | 2.2* | 6.1* | 1.2* | 0.8* | 2.2 | 5.8 |
| + TRACE | **9.4** | **18.2** | **8.2** | **13.1** | **7.8** | **11.2** | **2.5** | **4.7** | **1.2** | **2.6** | **4.3** | **9.5** | **2.0** | **1.6** | **5.1** | **8.7** |

\* Reproduced results using publicly accessible code for a fair comparison. The rest are the values reported in the publication.

ciency, while entropy- and Wasserstein-based alternatives underperform and the mutual-information variant (JSD) yields similar AP at higher latency.

**Distributional Consistency of $t^{\star}$.** To further characterize the distributional patterns of IEP (Sec. 3.2) beyond Fig. 7(b), we extend the analysis to different datasets and object categories, as shown in Fig. 14. Figure 14(a) reproduces the VOC 2012 train-set distribution of optimal timesteps $t^{\star}$ used in Fig. 7(b), while Fig. 14(b) shows the corresponding distribution on the much larger COCO 2014 train set (82,783 images). Across all five diffusion backbones, the COCO histograms closely match those from VOC, with $t^{\star}$ consistently concentrating in the same instance-aware range, indicating that the semantic-to-instance transition is stable across datasets and scales. In Fig. 14(c), we further restrict the analysis of IEP to images containing two most frequent classes in COCO and plot the per-class distributions of $t^{\star}$. Although there are mild class-dependent shifts, the instance-aware timesteps for all four classes lie predominantly in the noise-like instance regime rather than the semantic regime, and they remain largely concentrated between 30–60% of the diffusion trajectory, consistent with the global patterns observed in Fig. 14(b).

### E.3 EXTENDED BENCHMARKING ON DIVERSE DOMAINS

**Generalization to Diverse UIS Benchmarks.** To rigorously evaluate the generalization capability of TRACE across varied domains, we extend our unsupervised instance segmentation (UIS) experiments to seven benchmarks: COCO 2014 (Lin et al., 2014), COCO 2017 (Caesar et al., 2018), LVIS (Gupta et al., 2019), KITTI (Geiger et al., 2012), Objects365 (Shao et al., 2019), SA-1B (Kirillov et al., 2023), and VOC 2012 (Everingham et al., 2010). We compare TRACE (with SD3.5-L backbone (Esser et al., 2024)) against three representative UIS methods: TokenCut (Wang et al., 2023b), MaskCut (Wang et al., 2023a), and ProMerge (Li & Shin, 2024). For fair comparison, we strictly reproduce these baselines using their official public checkpoints and evaluation protocols (see Appendix F for details). Crucially, to align with the standard UIS protocol where models (Wang et al., 2023b;a) are typically fine-tuned on the ImageNet train set (Krizhevsky et al., 2012), we also fine-tune our diffusion backbone and edge generator on ImageNet. This allows us to generate TRACE-based instance edges using the ImageNet-trained model and apply them directly to downstream datasets without any target-specific adaptation. As shown in Tab. 12, TRACE consistently outperforms all baselines across all datasets, achieving an average $AP^{mk}$ of **5.1**, which corresponds to a **2.3× improvement** over the strongest baseline (ProMerge). Notably, on the challenging LVIS dataset (Gupta et al., 2019), which features a long-tail distribution, TRACE more than doubles the performance ($1.1 \rightarrow 2.5$ $AP^{mk}$), indicating superior handling of rare and diverse objects. Similarly, in the dense scenes of Objects365 (Shao et al., 2019), TRACE achieves a significant boost ($1.7 \rightarrow 4.3$ $AP^{mk}$), proving its efficacy in separating heavily occluded instances where traditional feature clustering often fails. These results confirm that the instance-aware cues extracted from diffusion priors capture fundamental structural boundaries rather than dataset-specific semantics. Consequently, this demonstrates that the refined instance edges produced by TRACE are robust and generalizable, effectively bridging the gap between semantic grouping and instance separation even in zero-shot transfer scenarios.

**Enhancement of Open-Vocabulary Segmentation Frameworks.** While most of our experiments focus on closed-set benchmarks, we additionally evaluate TRACE in open-vocabulary segmentation settings by attaching it to recent open-vocabulary models. Specifically, we integrate TRACE with TTD (Jo et al., 2024b) and Talk2DINO (Barsellotti et al., 2025) and measure performance on five standard open-vocabulary segmentation benchmarks: VOC 2012 (Everingham et al., 2010), Pascal Context

Table 13: **Performance of open-vocabulary segmentation.** We attach TRACE to two open-vocabulary segmentation methods and evaluate on five benchmarks.

| Method | VOC | Context | Stuff | ADE | City |
|---|---|---|---|---|---|
| TTD | 61.1 | 37.4 | 23.7 | 17.0 | 27.9 |
| + TRACE | **64.9** | **40.3** | **24.3** | **18.3** | **31.3** |
| Talk2DINO | 65.8 | 42.4 | 30.2 | 22.5 | 38.1 |
| + TRACE | **68.4** | **44.8** | **32.1** | **23.9** | **40.2** |

(Mottaghi et al., 2014), COCO-Stuff (Caesar et al., 2018), ADE20K (Zhou et al., 2019), and Cityscapes (Cordts et al., 2016). As shown in Tab. 13, refined instance boundaries from TRACE consistently improve open-vocabulary segmentation quality. On top of TTD (Jo et al., 2024b), TRACE yields gains of 0.6–3.8 points across datasets (*e.g.*, from 61.1 to 64.9 on VOC and from 27.9 to 31.3 on Cityscapes), and on top of Talk2DINO (Barsellotti et al., 2025) it further improves performance by 1.4–2.6 points (*e.g.*, from 65.8 to 68.4 on VOC and from 38.1 to 40.2 on Cityscapes). These results indicate that the instance-aware cues extracted by TRACE transfer beyond closed-set UIS and WPS (Tabs. 1 and 2) and provide consistent benefits for open-vocabulary segmentation, including real-world scenes, by sharpening object boundaries while preserving the underlying open-vocabulary recognition capability of the backbone models.

### E.4 LIMITATIONS AND FUTURE DIRECTIONS

**Challenges in Satellite Imagery (Tiny Instances).** While TRACE substantially improves instance segmentation on natural-image benchmarks (see Tabs. 1 and 2), we observe a clear limitation on datasets dominated by very small objects (*i.e.*, instances occupying only about $0.01\%$ of the image area). Table 14 reports results on two satellite benchmarks such as HRSID (Wei et al., 2020) and iSAID (Waqas Zamir et al., 2019), evaluated on their official test sets using $AP^{mask}$. When added on top of Mask R-CNN (ResNet-101+FPN) (He et al., 2017), TRACE

Table 14: **Performance on satellite benchmarks.** For a fair comparison, we evaluate instance segmentation performance on HRSID and iSAID test sets in terms of $AP^{mask}$.

| Method | HRSID | iSAID |
|---|---|---|
| Mask R-CNN | 65.4 | 25.6 |
| + TRACE | 50.3 (-5.1) | 20.2 (-5.4) |

leads to a degradation of 5.1 $AP^{mask}$ on HRSID (65.4 $\rightarrow$ 50.3) and 5.4 $AP^{mask}$ on iSAID (25.6 $\rightarrow$ 20.2). We attribute this failure mode to the resolution loss inherent in latent diffusion models: before denoising, all images are encoded by a VAE into a low-resolution latent grid (up to a $16\times$ spatial downsampling), as shown in Fig. 4, which severely compresses tiny structures. As a consequence, closely packed small objects tend to share blurred or merged boundaries in latent space, and the decoded instance-edge cues from TRACE cannot reliably separate individual instances in high-density satellite scenes. Qualitatively (see Fig. 15(a)), we often observe multiple nearby targets fused into a single instance mask. Addressing this limitation likely requires diffusion backbones with higher-resolution latents or hybrid schemes that combine TRACE with high-resolution, task-specific feature extractors for small-object regimes.

**Applicability to Medical Imaging (Out-of-Distribution).** TRACE is built on text-to-image diffusion models trained on natural images, which raises concerns about its behavior on out-of-distribution domains such as histopathology images. In Tab. 15, we evaluate two cell instance segmentation methods, SSA (Sahasrabudhe et al., 2020) and COIN (Jo et al., 2025), on the MoNuSeg (Kumar et al., 2020) and TNBC (Naylor et al., 2019) test sets using PQ. Adding TRACE on top of these backbones consistently degrades performance: PQ

Table 15: **Performance on medical benchmarks.** For a fair comparison, we evaluate panoptic segmentation performance on MoNuSeg/TNBC test sets in terms of PQ.

| Method | MoNuSeg | TNBC |
|---|---|---|
| SSA | 0.185 | 0.253 |
| + TRACE | 0.148 (-0.037) | 0.209 (-0.044) |
| COIN | 0.536 | 0.540 |
| + TRACE | 0.439 (-0.097) | 0.426 (-0.114) |

drops from 0.185 to 0.148 on MoNuSeg and from 0.253 to 0.209 on TNBC for SSA, and from 0.536 to 0.439 (MoNuSeg) and 0.540 to 0.426 (TNBC) for COIN. We hypothesize that this limitation stems from a domain mismatch between natural-image diffusion priors and medical imagery: the diffusion backbones we use are trained on photographs, not histopathology slides, and their latent representations tend to emphasize color and texture patterns that do not align with cell boundaries. Consequently, incomplete instance edges predicted by TRACE often undersegment cells, as seen

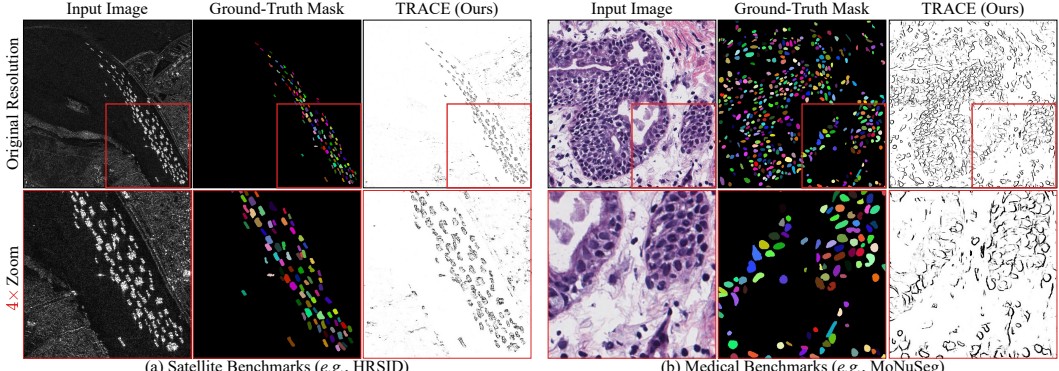

Figure 15: **Failure cases on satellite and medical benchmarks.** Two examples illustrate key limitations of TRACE on tiny objects and out-of-distribution domains.

in Fig. 15(b). These results indicate that directly applying TRACE to highly out-of-distribution medical images can be harmful, and suggest that future extensions should either adapt the diffusion prior to the medical domain (*e.g.*, via domain-specific diffusion training) or combine TRACE with medical-specific feature extractors and supervision.

## F    DATASETS AND METRICS FOR EVALUATION

**Datasets for Unsupervised Instance Segmentation.** We evaluate our TRACE on seven benchmarks including COCO2014 (Lin et al., 2014) and COCO2017 (Caesar et al., 2018), LVIS (Gupta et al., 2019), KITTI (Geiger et al., 2012), Objects365 (Shao et al., 2019), and SA-1B (Kirillov et al., 2023). COCO2014 (Lin et al., 2014) and COCO2017 (Caesar et al., 2018) are standard datasets for object detection and segmentation. COCO2014 has 80 classes, 83K training images and 41K validation images. COCO2017 is composed of 118K and 5K images for training and validation splits respectively. For results on COCO2014 and COCO2017, we use the images in the validation split. LVIS has densely-annotated instance masks, making it more challenging for segmentation. We test our performance on the validation set containing 245K instances on 20K images. For KITTI and Objects365, we evaluate on 7K images and a subset of 44K images in the val split, respectively. Lastly, for SA-1B, we assess on a subset of 11K images, which come with 100+ annotations per image on average.

**Datasets for Weakly-supervised Panoptic Segmentation.** Pascal VOC (Everingham et al., 2010) consists of 20 ”thing” and 1 ”stuff” categories. It contains 11K images for training and 1.4K images for validation. COCO2017 (Caesar et al., 2018) has 80 ”thing” and 53 ”stuff” categories, which is a challenging benchmark for PS. We validate TRACE on the 5K images in the validation dataset.

**Datasets for Instance Edge Evaluation.** To directly evaluate instance-edge quality, we construct an instance-aware edge benchmark from the COCO 2014 validation set (Lin et al., 2014). For each of the 41K validation images, we start from the ground-truth panoptic segmentation masks and extract instance boundaries by labeling pixels that lie on the borders between distinct panoptic segments as positive edges, and all remaining pixels as background, as shown in Fig. 11. The resulting binary instance-edge maps are used as ground truth for all our edge-quality experiments (ODS/OIS and connectivity metrics (Shit et al., 2021)) on real images, enabling a direct comparison between TRACE and conventional edge detectors Xie & Tu (2015); Su et al. (2023) under an instance-aware boundary supervision rather than generic low-level contour annotations.

**Evaluation Metrics for Edge Quality.** To evaluate the quality of instance boundaries, we follow the standard evaluation protocols widely adopted in general edge detection methods such as HED (Xie & Tu, 2015) and PiDiNet (Su et al., 2023). We report the standard F1-score (or F-measure, defined as $\frac{2 \cdot \text{Precision} \cdot \text{Recall}}{\text{Precision} + \text{Recall}}$) using two standard metrics: Optimal Dataset Scale (ODS) and Optimal Image Scale (OIS). ODS computes the F1-score using a global threshold that is optimal across the entire test dataset, providing a measure of generalizability. OIS selects the optimal threshold for each individual image to maximize the F1-score, reflecting the best possible performance per image.

Furthermore, since pixel-wise metrics often fail to capture the topological correctness of thin boundary structures, we additionally employ the clDice (centerline Dice) metric (Shit et al., 2021). Unlike the standard Dice coefficient, clDice calculates the overlap between the skeletons (centerlines) of the predicted edges and the ground truth. This allows for a more robust evaluation of topological connectivity and structural preservation of the instance edges.

**Evaluation Metrics for Instance Segmentation.** We evaluate the performance comparison of UIS methods with and without TRACE based on average precision (AP) and average recall (AR) on Pascal VOC and MS COCO dataset. Precision and Recall are defined as

$$\text{Precision} = \frac{\text{TP}}{\text{TP} + \text{FP}}, \ \text{Recall} = \frac{\text{TP}}{\text{TP} + \text{FN}} \tag{17}$$

where TP, FN, FP are short for True Positive, False Negative, and False Positive, respectively. Intuitively, a high precision means the method has a low rate of making false positives, but it does not imply that all the positives were found. A high recall means the method has a low rate of making false negatives.

AP measures the precision across different recall levels. A high AP means the method finds most objects(recall) while minimizing false positives (precision). On COCO, AP is calculated over different IoU thresholds and object sizes, then averaged. For Pascal VOC, mAP is used, averaging AP across classes at a single IoU threshold (0.5). COCO uses mAP@IoU=0.5:0.95, meaning it averages AP across ten IoU thresholds(0.50, 0.55, ..., 0.95).

$$AP = \int_0^1 \text{precision}(r)dr \tag{18}$$

$$mAP_{\text{COCO}} = \frac{mAP_{0.50} + mAP_{0.55} + \cdots + mAP_{0.95}}{10} \tag{19}$$

Average Recall (AR) is calculated as the area under the recall-threshold curve. Like AP, it is averaged over multiple IoU thresholds. AR reflects how well the method recalls true objects rather than balancing precision and recall. In COCO, AR is reported as AR@100, AR@100 (small), AR@100 (medium), and AR@100 (large), corresponding to the average recall with up to 100 detections per image across different object sizes.

**Evaluation Metrics for Panoptic Segmentation.** We report evaluation results on the standard evaluation metrics of panoptic segmentation task, including panoptic quality (PQ), segmentation quality (SQ) and recognition quality (RQ). PQ is defined for matched segments (IoU > 0.5 between predicted and ground truth masks) and combines SQ and RQ:

$$PQ = \frac{\sum_{(p,gt)\in TP} \text{IoU}(p, gt)}{|TP| + 0.5|FP| + 0.5|FN|} \tag{20}$$

Segmentation quality (SQ) measures the accuracy of segment boundaries, focusing only on segments that were correctly identified and ignoring false positives and false negatives. SQ is defined as:

$$SQ = \frac{\sum_{(p,gt)\in TP} \text{IoU}(p, gt)}{|TP|} \tag{21}$$

Recognition quality (RQ) measures the model's ability to correctly classify instances, accounting for both precision and recall for detected instances. RQ combines recall (finding all objects) with precision (only detecting true objects), focusing on instance recognition rather than segmentation quality. RQ is defined as:

$$RQ = \frac{|TP|}{|TP| + 0.5|FP| + 0.5|FN|} \tag{22}$$

Table 16: Evaluation Segmentation Metrics on Pascal VOC and COCO datasets.

| Metric | Intuitive Meaning |
|--------|-------------------|
| AP | Balance between precision and recall |
| AR | Ability to recall true objects |
| PQ | Combined segmentation quality and recognition quality |
| SQ | Accuracy of segmentation shapes |
| RQ | Correct classification of instances |

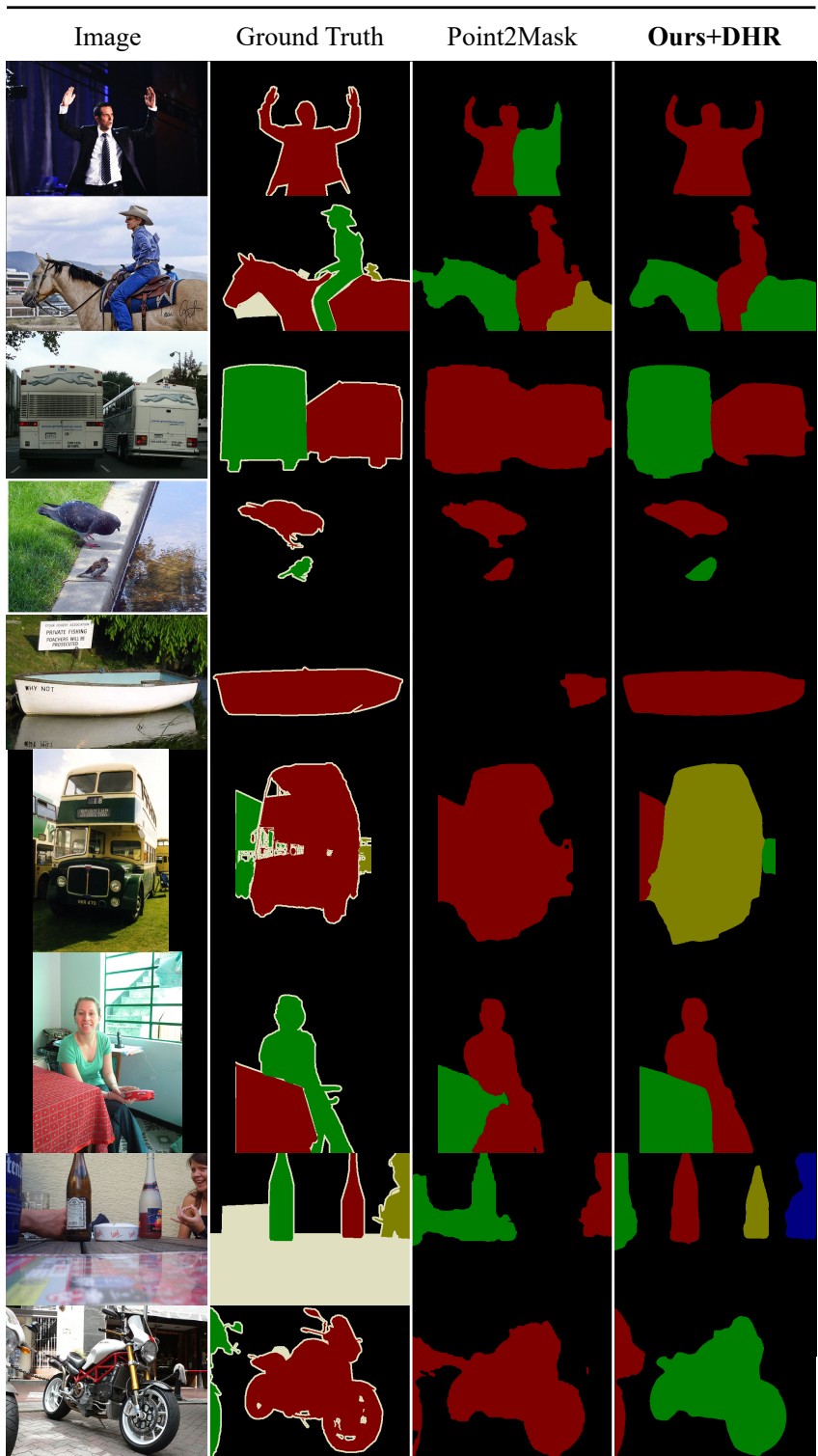

Figure 16: **Qualitative results in WPS with ours (TRACE+DHR (Jo et al., 2024a)) and point-supervised Point2Mask (Li et al., 2023b) on the VOC2012 (Everingham et al., 2010) validation set.** (Ours) We trained a Mask2Former using a ResNet-50 backbone with pseudo panoptic masks generated from TRACE+DHR. The samples in this figure are the outputs from Mask2Former trained with our masks.

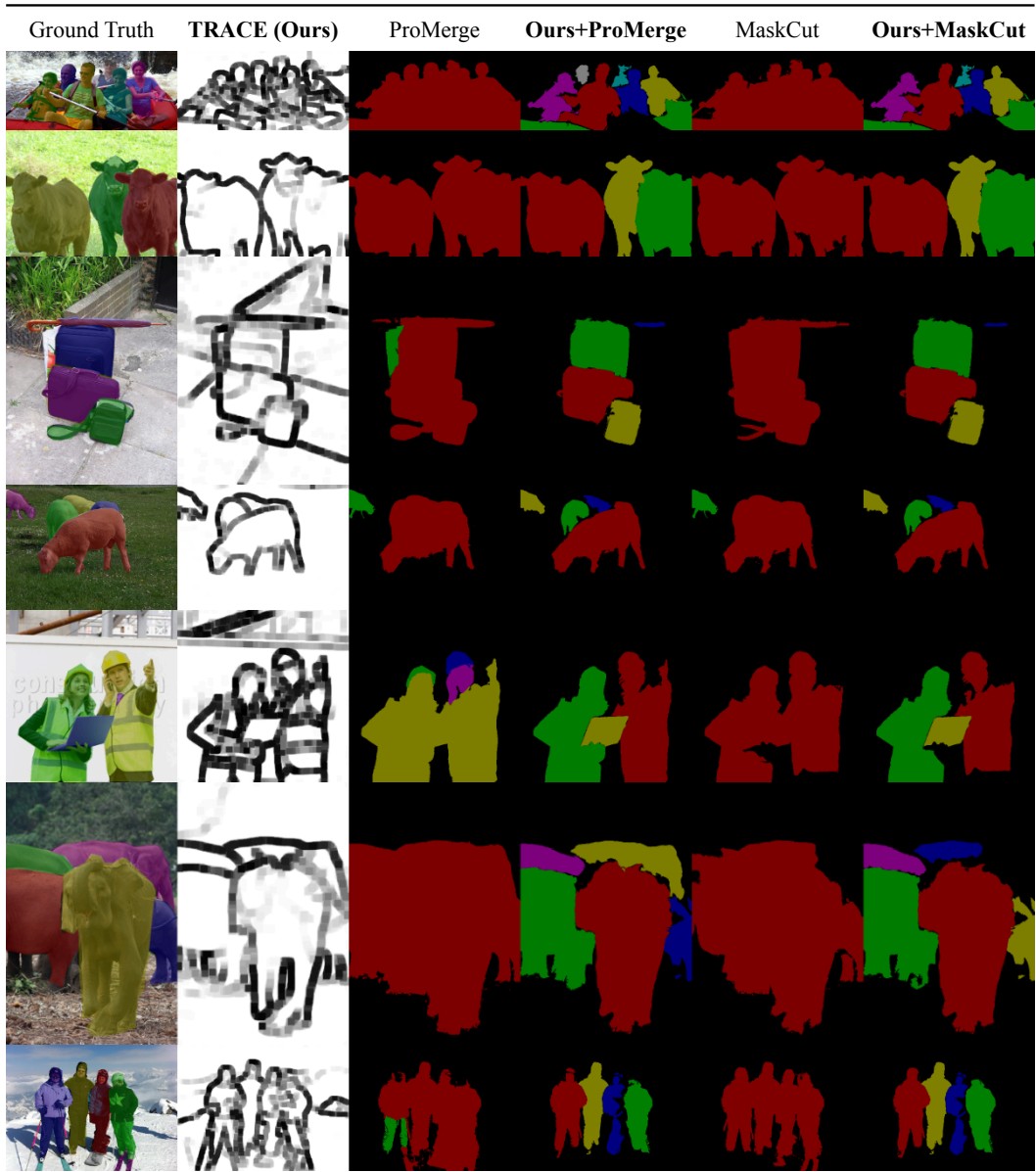

Figure 17: Qualitative results in UIS with ours (TRACE+ProMerge, TRACE+MaskCut) and existing methods (ProMerge (Li & Shin, 2024), MaskCut (Wang et al., 2023a)) on the COCO2014 (Lin et al., 2014) validation set.

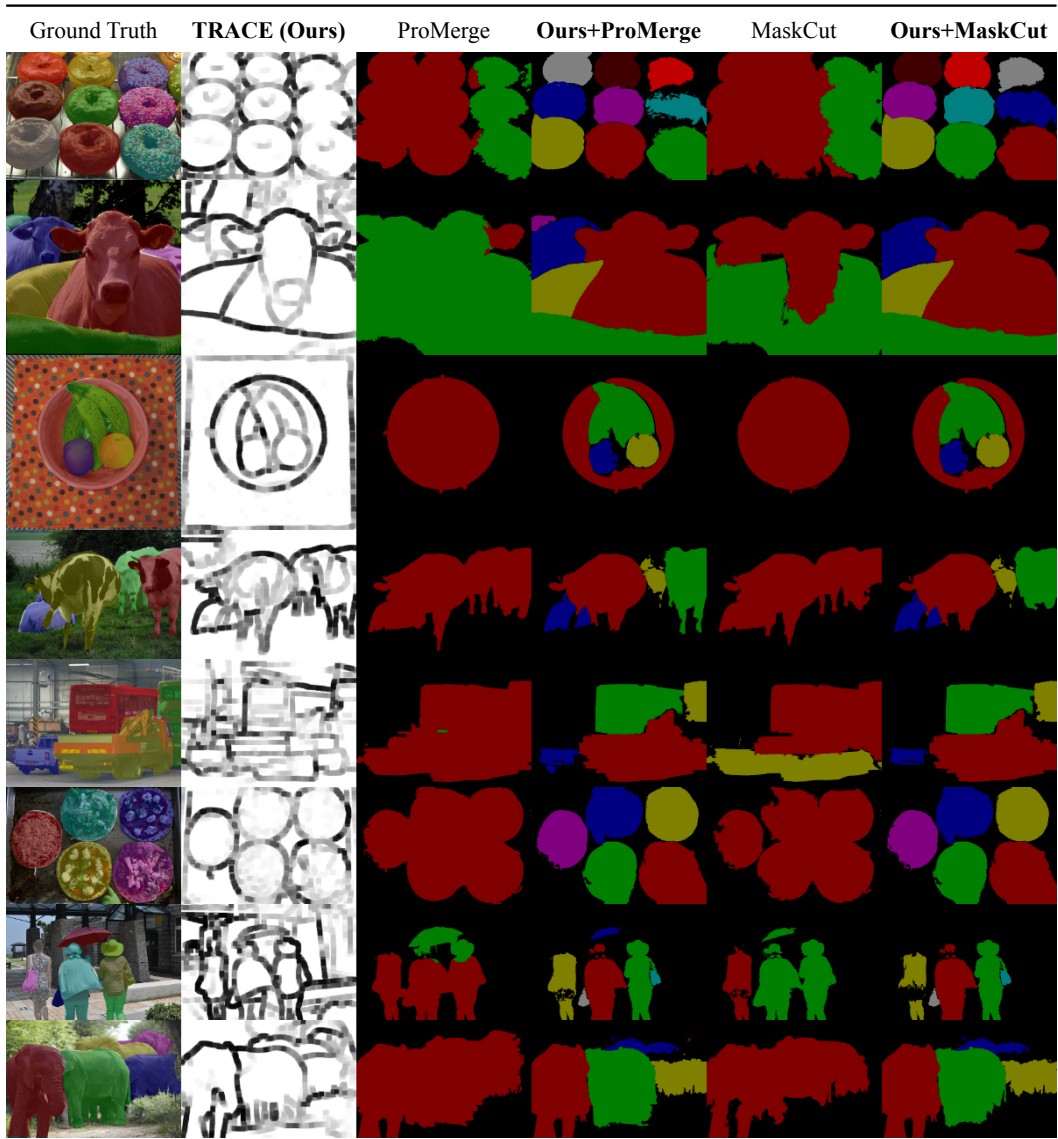

Figure 18: More Qualitative results in UIS with ours (TRACE+ProMerge, TRACE+MaskCut) and existing methods (ProMerge (Li & Shin, 2024), MaskCut (Wang et al., 2023a)) on the COCO2014 (Lin et al., 2014) validation set.

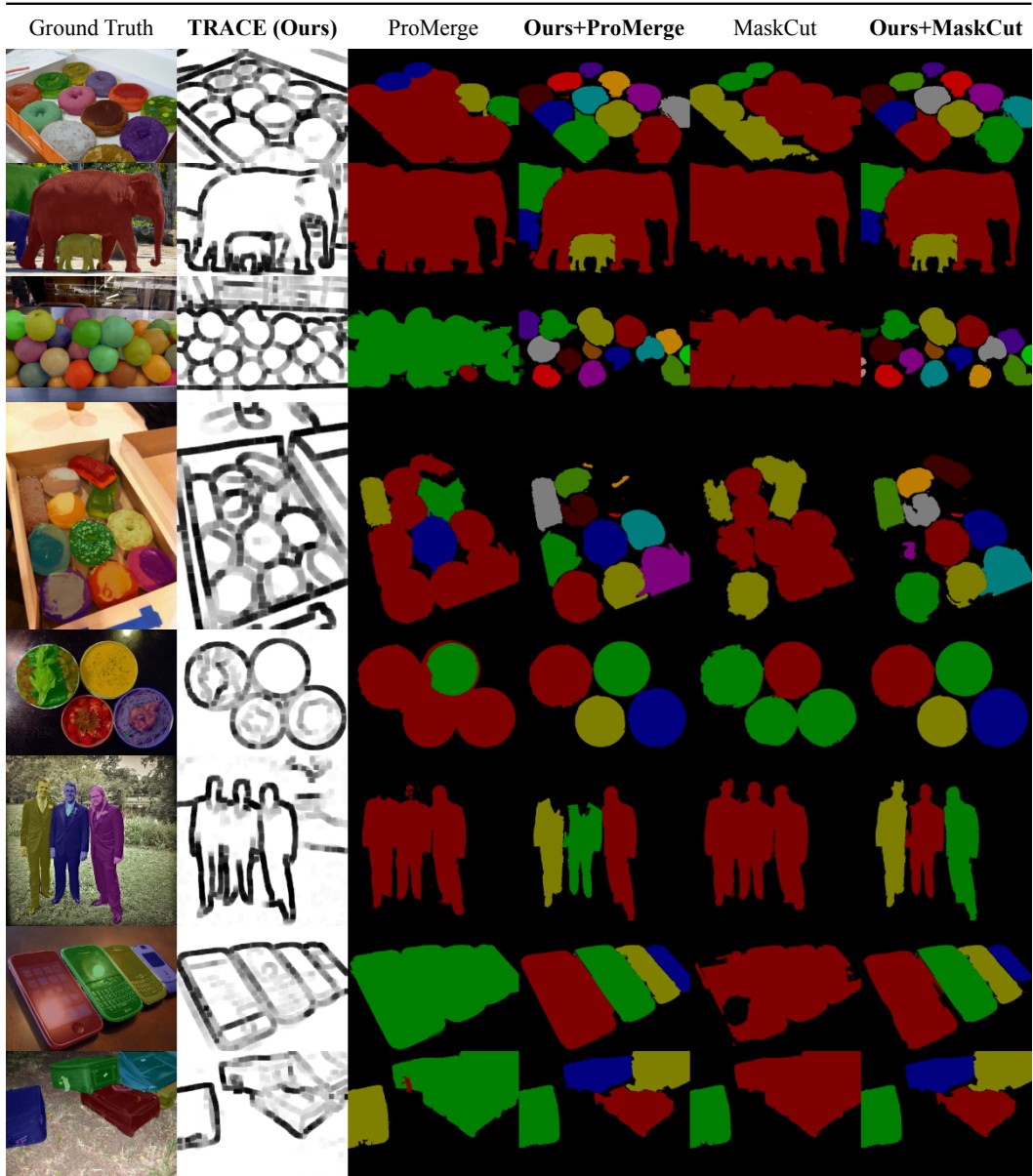

Figure 19: More Qualitative results in UIS with ours (TRACE+ProMerge, TRACE+MaskCut) and existing methods (ProMerge (Li & Shin, 2024), MaskCut (Wang et al., 2023a)) on the COCO2014 (Lin et al., 2014) validation set.

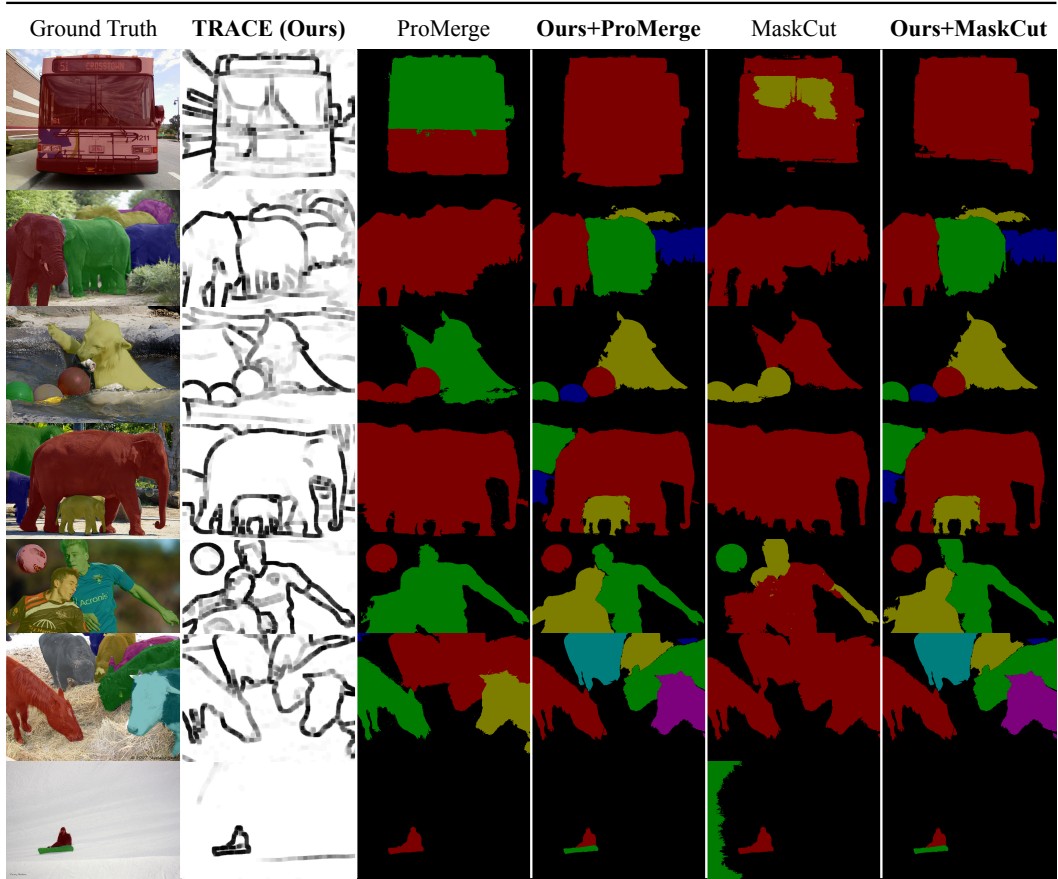

Figure 20: Qualitative results in UIS with ours (TRACE+ProMerge, TRACE+MaskCut) and existing methods (ProMerge (Li & Shin, 2024), MaskCut (Wang et al., 2023a)) on the COCO2017 (Caesar et al., 2018) validation set.

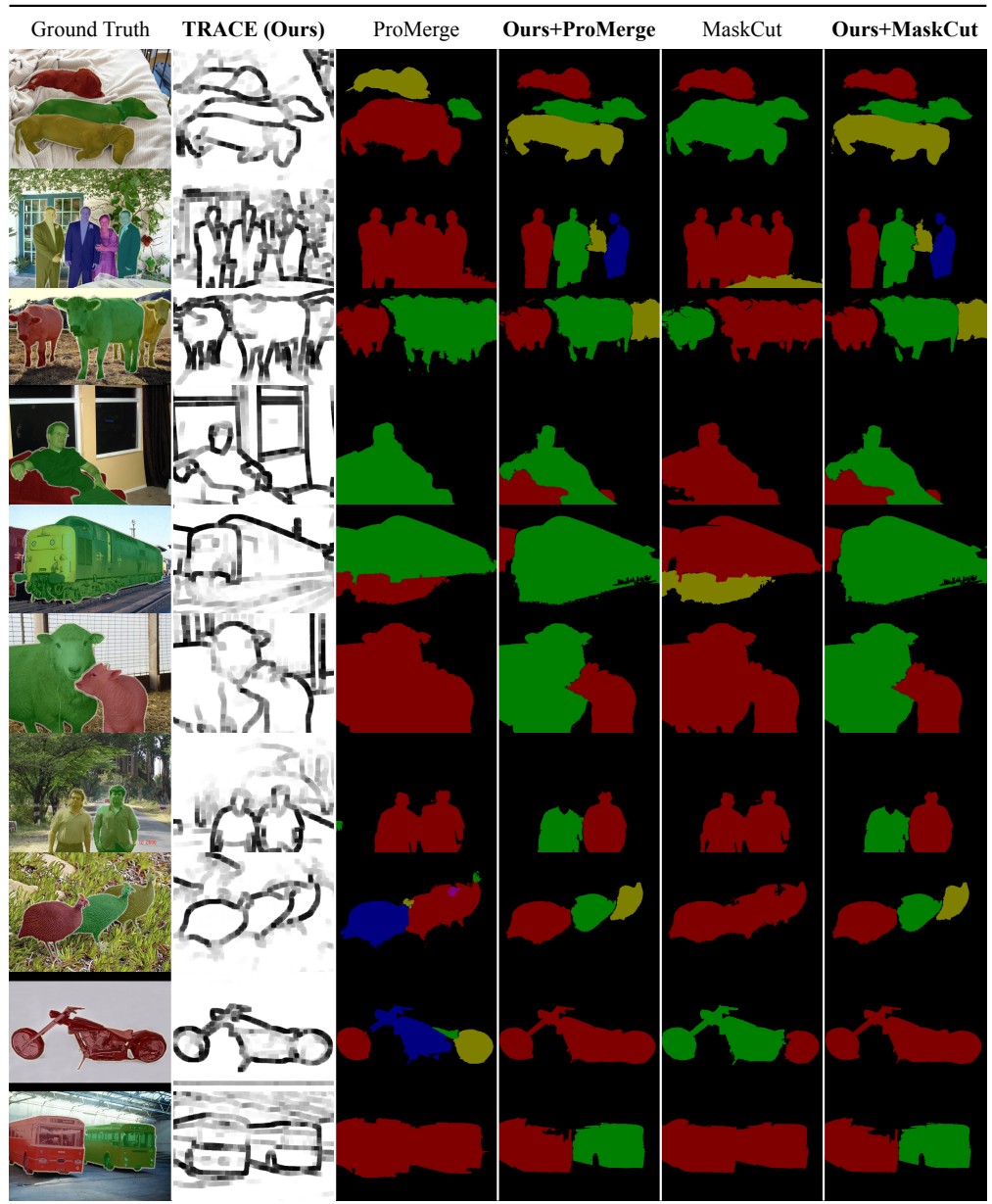

Figure 21: Qualitative results in UIS with ours (TRACE+ProMerge, TRACE+MaskCut) and existing methods (ProMerge (Li & Shin, 2024), MaskCut (Wang et al., 2023a)) on the VOC2012 (Everingham et al., 2010) validation set.

