# OpenReview forum: "TRACE: Your Diffusion Model is Secretly an Instance Edge Detector"
_ICLR.cc/2026/Conference — ICLR 2026 Oral_

### Official Review · Reviewer_JkD6 · 2025-10-28

**Soundness:** 3
**Presentation:** 3
**Contribution:** 3
**Rating:** 6
**Confidence:** 5

**Summary:**

TRACE shows that text-to-image diffusion models already encode sharp instance boundaries in their self-attention maps at a special denoising step (the Instance Emergence Point, IEP). By harvesting these “secret” edges with a simple, KL-based Attention Boundary Divergence (ABDiv) and distilling them into a one-step edge decoder, TRACE delivers: +5.1 AP on unsupervised COCO instance segmentation, +1.7 PQ on tag-supervised panoptic segmentation (no boxes/masks), 81× faster than per-image diffusion inversion
All without a single manual instance label.

**Strengths:**

1. First to treat diffusion self-attention as a free instance-edge oracle.
2. Introduces IEP + ABDiv, two lightweight, non-parametric cues that any diffusion backbone yields.
3. Exhaustive ablations (Tab. 3, 4, 5) and 5 backbones; KL beats MSE/MAE by > 5 AP.
4. Clear pipeline figure (Fig. 4); every symbol defined; 81× speed-up quantified.
5. Annotation-free edge seeds outperform point-supervised panoptic models (Tab. 2).

**Weaknesses:**

1. Only one 1-layer CNN decoder is tried; U-Net, light transformer, or multi-scale may sharper edges.
2. Paper shows success cases (Fig. 2, 9); no figure of merged/fragmented predictions when IEP fails.
3. TRACE claims “no prompt needed”, but null-text vs. descriptive prompt may shift t* or edge quality.

**Questions:**

see weakness

---

> ### Author Response · Authors · 2025-11-25
> **Response to Reviewer JkD6**
>
> We sincerely appreciate Reviewer JkD6 for the positive assessment and for recognizing the novelty of treating diffusion self-attention as a free instance-edge oracle. We are particularly encouraged that you found our exhaustive ablations and clear presentation to be strong points. Below, we address your concerns with new experiments and analyses added to our revision.
>
> **[W1] Effect of Edge Decoder Capacity (vs. U-Net/Mask2Former)**
>
> We agree that exploring stronger edge decoders is essential to verify if the limitation lies in our lightweight design. Following your suggestion, we replaced our lightweight decoder (0.1 MB) with heavier architectures: a U-Net style decoder (34 MB) and a Mask2Former-style decoder (258 MB), keeping the same diffusion backbone (SD3.5-L) and setup (COCO 2014). As shown in **Table R1**, increasing the decoder capacity by up to $2,580\times$ yields negligible performance gains ($+0.0 \sim +0.2$ $AP^{mk}$).
>
> **Table R1: Effect of edge decoder capacity on COCO 2014.**
> | Method | Params. of Edge Decoder | $AP^{mk}$ (Gain) |
> | :--- | :---: | :---: |
> | ProMerge (Baseline) | - | 3.1 |
> | + TRACE (Ours; Default) | **0.1 MB** | **8.2 ($2.6\times$)** |
> | + TRACE (U-Net) | 34 MB | 8.2 ($2.6\times$) |
> | + TRACE (Mask2Former) | 258 MB | 8.4 ($2.7\times$) |
>
> These results confirm two key points:
>
> 1. The instance-aware signals extracted from text-to-image diffusion models via TRACE are already precise; a simple linear projection is sufficient to decode them.
>
> 2. As detailed in Sec. 3.1, we aggregate self-attention maps from all layers. Thus, multi-scale information is already inherently integrated before the decoding stage, reducing the need for complex multi-scale decoders like Mask2Former.
>
> **[W2] Limitations**
>
> Thank you for this valuable suggestion. While TRACE shows robust improvements across 11 real-world benchmarks (see Tabs. 1/2/12/13), we have identified and analyzed two primary failure cases. We have added a detailed discussion in Appendix E.4, quantitative results in Tabs. 14/15, and qualitative visualizations in Fig. 15.
>
> * Tiny Objects (Satellite Imagery): On datasets like HRSID and iSAID, where instances occupy $\approx 0.01$\% of the image area, TRACE lags behind the baseline by $\approx 5$\%. This is due to the spatial compression (/8 or /16) of the VAE in latent diffusion models (i.e., all diffusion backbones for TRACE), which blurs boundaries of extremely densely packed, tiny objects.
>
> * Domain Shift (Medical Imagery): On MoNuSeg and TNBC (histopathology), performance drops by at least 3\%. Since the diffusion backbone is pretrained on natural images, its internal self-attention maps miscapture cell boundaries in medical domains.
>
> We believe explicitly stating these limitations clarifies the operating scope of TRACE (natural images compatible with generative diffusion priors) and suggests clear future directions (e.g., domain-specific fine-tuning).
>
> **[W3] Dependence on Prompts (Null-text vs. Descriptive)**
>
> To address your concern about whether descriptive prompts might shift the IEP $t^\star$ or improve edge quality, we conducted an ablation study using instance-aware prompts constructed from ground-truth instance annotations (e.g., "A photo of [N] [class]"). As shown in **Table R2** (and discussed in Appendix E.1), using descriptive prompts yields only a marginal gain compared to our default null-text setting.
>
> **Table R2. Effect of prompt types on COCO 2014.**
> | Prompt Type | Description | Example | $AP^{mk}$ |
> | :--- | :--- | :--- | :---: |
> | Null-text (Ours; Default) | Empty string | "" | 8.2 |
> | Instance-aware | "A photo of [N] [class]" | "A photo of two cats" | 8.3 (+0.1) |
>
> Furthermore, we observed that the distribution of the optimal timestep $t^*$ remains largely unchanged. This strongly supports our claim that instance boundaries are intrinsically encoded in the diffusion process itself, rather than being driven by the text condition. TRACE effectively harvests these "secret" cues without requiring prompt engineering.
>
> We hope these additional experiments and the new limitations section fully address your concerns. We are happy to answer any further questions.

---

> > ### Comment · Reviewer_JkD6 · 2025-11-27
> >
> > Thanks for your response. The response addresses my concerns on the experimental details. I will keep my score.

---

> > > ### Author Response · Authors · 2025-11-27
> > >
> > > We deeply thank you for confirming that our response addressed your concerns regarding the experimental details. To further contribute to the research community and ensure full reproducibility, we are preparing to release our complete codebase, incorporating the scripts for all new benchmarks and ablation studies conducted during this rebuttal.

---

### Official Review · Reviewer_KrZ9 · 2025-10-28

**Soundness:** 3
**Presentation:** 4
**Contribution:** 4
**Rating:** 8
**Confidence:** 4

**Summary:**

This paper proposes **TRACE** (TRAnsforming diffusion Cues to instance Edges), a novel framework revealing that diffusion models inherently encode *instance-level boundary information* within their self-attention maps.
The authors identify the Instance Emergence Point (IEP)—the timestep where object structures first appear—and introduce Attention Boundary Divergence (ABDiv) to extract instance edges without any annotation.
Through one-step edge distillation, TRACE achieves real-time inference and demonstrates strong results on unsupervised and weakly-supervised segmentation tasks.

**Strengths:**

1. **Novel Insight** – The discovery that diffusion models internally encode instance boundaries is original and insightful.
2. **Elegant Design** – The approach leverages self-attention statistics (KL divergence) without requiring extra supervision or training labels.
3. **Strong Empirical Results** – TRACE significantly outperforms state-of-the-art methods like MaskCut and ProMerge on COCO and VOC datasets.
4. **Interpretability** – The visualization of attention evolution (IEP phenomenon) provides an interpretable window into diffusion model internals.
5. **Efficiency** – The one-step distillation achieves ~81× inference acceleration, demonstrating practical usability.

**Weaknesses:**

1. **Limited Theoretical Explanation** – While IEP is clearly defined using KL divergence peaks, the underlying cause of such emergence remains described qualitatively. A lightweight theoretical or statistical discussion (e.g., how noise levels influence attention entropy) could enhance the argument.

2. **Negative Analysis** – The paper focuses on successful examples; adding a few failure visualizations or robustness analyses (e.g., under cluttered scenes or overlapping objects) would provide a more balanced evaluation.

3. **Fine-Grained Analysis of IEP** – The phenomenon of IEP is intriguing, but its quantitative behavior (e.g., variance across images, layers, or diffusion steps) is not deeply examined. A more detailed statistical characterization—such as distributional patterns or consistency metrics—could enrich understanding and further substantiate the discovery.

**Questions:**

1. How consistent is the identified IEP across diffusion timesteps, model scales, or datasets?  For example, does the same timestep correspond to “edge emergence” across all classes and diffusion models?
2. Could the KL divergence criterion for IEP detection be replaced or augmented by other measures (e.g., entropy, mutual information)?
3. How sensitive is the distilled edge decoder to the choice of teacher diffusion backbone? Could it generalize across unseen models or domains (e.g., medical, satellite images)?

---

> ### Author Response · Authors · 2025-11-25
> **[1/2] Response to Reviewer KrZ9**
>
> We are delighted to receive such an enthusiastic assessment from Reviewer KrZ9. It is highly encouraging that you identified the "novel insight" and "elegant design" as key strengths of our work. We also appreciate your recognition of TRACE's interpretability and efficiency. Your constructive suggestions regarding theoretical grounding and failure analysis have significantly enriched our revision. Below, we address your questions and weaknesses point-by-point.
>
> **[Q1 & W3] Consistency and Robustness of IEP**
>
> To address your query on whether the IEP is a stable phenomenon, we conducted extensive analyses across diffusion backbones, datasets, and random seeds. We have expanded our analysis in Appendix E.2 and added Fig. 14.
>
> * Consistency across Datasets and Models: As shown in the new Fig. 15, we computed IEP histograms for 5 different diffusion backbones on both VOC 2012 and COCO 2014 train sets. Across all models and datasets, the optimal timestep $t^*$ consistently emerges in the 30–60% denoising range (instance regime; green), clearly distinct from the semantic regime (red). For a fixed dataset (COCO) and backbone (SD3.5-L), we varied the random seed for the inversion noise 10 times. The resulting IEP histogram remained virtually identical, confirming that IEP is a property of the image structure, not random noise artifacts.
> * Step Size: While a smaller step size (e.g., 25) yields marginally better edge precision, Figure 7(a) and Table 5 confirm that our chosen step size of 100 provides the optimal trade-off, maintaining high performance across all 5 backbones while keeping latency low.
>
> **[Q2] Alternatives to KL Divergence for IEP**
>
> You asked if other measures like entropy or mutual information could replace KL. We have expanded Tab. 4 into a comprehensive comparison (**Table R1**, added to Appendix E.2 as Table 11) involving 6 metrics on both VOC 2012 and COCO 2014.
>
> **Table R1. Comparison of similarity metrics for IEP.**
> | Metric | Latency/img | VOC ($AP^{mk}$) | COCO ($AP^{mk}$) |
> | :--- | :---: | :---: | :---: |
> | KL (Ours; Default) | 3,082 ms | **9.4** | **8.2** |
> | JSD | 5,120 ms | 9.4 | 8.1 |
> | MSE (L2) | 1,232 ms | 3.8 | 3.5 |
> | MAE (L1) | 924 ms | 3.5 | 3.3 |
> | Entropy | 2,070 ms | 5.1 | 4.1 |
> | Wasserstein ($W_1$) | 3,434 ms | 6.2 | 5.0 |
>
> As a result, the KL divergence remains the most effective choice. While Jensen-Shannon Divergence (JSD), a proxy for mutual information, achieves similar accuracy, it is $\approx 1.7\times$ slower. Other metrics like Entropy or Wasserstein distance yield significantly lower AP ($5.0 \sim 6.2$). Thus, KL offers the best balance.
>
> **[Q3-1] Sensitivity to Teacher Diffusion Backbone**
>
> You asked about the sensitivity of the distilled edge decoder to the choice of the teacher diffusion backbone.
>
> First, we clarify that direct weight transfer of the edge decoder between different diffusion models is not feasible because the self-attention patterns and channel dimensions vary across architectures (e.g., SD1.5 vs. SD3.5-L). Therefore, we interpret your question as an inquiry into the robustness of the TRACE framework across different backbones.
>
> To verify this, we evaluated TRACE across 5 diffusion models and 5 non-diffusion foundation models. Note that for non-diffusion backbones (e.g., DINO, VLM), we applied ABDiv only directly to their self-attention maps, as they lack the temporal denoising trajectory required for IEP. For diffusion backbones, we used the full IEP + ABDiv pipeline.
>
> **Table R2. Effect of backbone choice on COCO 2014.**
> | Method | Backbone | Params. | $AP^{mk}$ | $AR^{mk}_{100}$ |
> | :--- | :--- | :---: | :---: | :---: |
> | ProMerge | -- | -- | 3.1 | 7.6 |
> | | **_Non-diffusion_** | | | |
> | + TRACE | DINOv2-G | 1.1B | 2.6 | 7.7 |
> | + TRACE | EVA02-E | 5.0B | 3.2 | 7.9 |
> | + TRACE | DINOv3 | 7.0B | 4.3 | 8.9 |
> | + TRACE | LLaVA | 13B | 3.8 | 8.4 |
> | + TRACE | Qwen2.5-VL | 72B | 4.1 | 8.5 |
> | | **_Diffusion_** | | | |
> | + TRACE | SD1.5 | 0.8B | 6.8 | 11.2 |
> | + TRACE | PixArt-$\alpha$ | **0.6B** | 7.1 | 11.8 |
> | + TRACE | SDXL | 2.5B | 7.4 | 12.3 |
> | + TRACE | SD3.5-L | 8.1B | 8.2 | 13.1 |
> | + TRACE | FLUX.1 | 12B | **8.3** | **13.4** |
>
> two key results are below:
> 1. Even the smallest diffusion model, PixArt-$\alpha$ (0.6B), achieves 7.1 $AP^{mk}$, more than doubling the baseline performance ($2.2\times$). In contrast, massive non-diffusion models like Qwen2.5-VL (72B) show only marginal gains ($1.3\times$, 4.1 $AP^{mk}$), failing to surpass even the smallest diffusion backbone.
> 2. Within diffusion models, performance steadily improves with model capacity (from 6.8 to 8.3 AP), demonstrating that TRACE effectively leverages the richer generative priors of stronger teachers.
>
> This confirms that our method is not sensitive to specific architectures but rather relies on the generative nature of diffusion models, which uniquely encodes instance-level boundaries that discriminative or VLM backbones lack.
>
> (Continued in Next Response)

---

> ### Author Response · Authors · 2025-11-25
> **[2/2] Response to Reviewer KrZ9**
>
> (Continued from Previous Response)
>
> **[W1] Theoretical Explanation of IEP**
>
> We appreciate your suggestion to provide a theoretical grounding for the IEP phenomenon. We have added Appendix D, which offers a lightweight information-theoretic derivation connecting diffusion noise levels to attention entropy.
>
> **Intuitive Explanation:** Imagine the attention mechanism as a "soft" selector controlled by a temperature parameter $\gamma_t$, which is tied to the Signal-to-Noise Ratio (SNR).
> 1. High Noise ($\gamma_t \approx 0$): The image is mostly noise. Attention is nearly uniform (high entropy), and small changes in noise level barely affect the distribution.
> 2. Low Noise (High $\gamma_t$): The image is clean. Attention is already sharply focused on target pixels (low entropy), and further denoising causes little change.
> 3. The "Sweet Spot" (IEP $t^\star$): In between, there is a transition phase where the attention distribution shifts most rapidly from uniform to focused. This is where the Fisher information with respect to $\gamma_t$ (and hence the noise level) peaks.
>
> **Formal derivation (Condensed from Appendix D):** Modeling attention rows as Gibbs distributions $p_t(j|i) \propto \exp(\gamma_t s_{ij})$, we derive that the rate of entropy change is proportional to the variance of similarities:
>
> $$\frac{dH_t}{dt} = -\gamma_t \frac{d\gamma_t}{dt} \operatorname{Var}_{p_t}[s]$$
>
> Furthermore, for small step sizes, the KL divergence between consecutive steps approximates the Fisher information of $\gamma_t$:
>
> $$\operatorname{KL}(p_{t-\Delta t} || p_t) \approx \frac{1}{2} (\Delta \gamma_t)^2 I(\gamma_t)$$
>
> where $I(\gamma_t) = \operatorname{Var}_{p_t}[s]$ is the Fisher information with respect to $\gamma_t$.
>
> This explains why IEP (the peak of KL) coincides with the moment of maximum structural emergence. It supports the point where the model's internal representation is most sensitive to the unveiling of the underlying signal.
>
> **[W2 & Q3-2] Failure Analysis and Generalization to Unseen Domains**
>
> To provide the balanced evaluation you requested, we looked beyond our success cases on COCO/VOC and tested TRACE on challenging out-of-distribution domains. We have added Fig. 15 and Tabs. 14/15 in Appendix E.4.
>
> While TRACE excels in natural scenes (robust improvements across 11 real-world benchmarks in Tabs. 1/2/12/13), we identified two clear limitations:
>
> 1. Tiny Objects (Satellite: HRSID, iSAID): Performance drops by ≈5% compared to baselines. The VAE compression in text-to-image diffusion models (i.e., our backbones) merges features of extremely small, dense/cluttered instances (occupying ≈0.01% area), making boundary extraction physically impossible at that resolution.
>
> 2. Unseen Domain (Medical: MoNuSeg, TNBC): Performance drops by ≈3%. Since our diffusion backbones are pretrained on natural images (LAION, etc.), their attention maps capture textures/edges relevant to natural objects, which misaligns with cell boundaries in histopathology.
>
> These results clarify that TRACE is best suited for natural images compatible with the diffusion prior, and suggests that domain-specific fine-tuning or higher-resolution diffusion models would be necessary for these specialized tasks.
>
> We hope these rigorous consistency checks, theoretical derivations, and transparent failure analyses fully address your comments. We are happy to engage in further discussion.

---

### Official Review · Reviewer_rCqE · 2025-10-30

**Soundness:** 3
**Presentation:** 3
**Contribution:** 3
**Rating:** 6
**Confidence:** 4

**Summary:**

The authors observe that self-attention in diffusion models encodes instance-level structure during denoising. Based on this observation, the authors propose to decode instance boundaries directly from pre-trained text-to-image diffusion models. This is achieved by identifying the Instance Emergence Point, extracting boundaries through Attention Boundary Divergence, and distilling them into a lightweight one-step edge decoder. Experimental results on unsupervised instance segmentation and weakly-supervised panoptic segmentation demonstrate the effectiveness of the proposed method.

**Strengths:**

-	The proposed method is well motivated. The authors start from the interesting observation that instance-level structure emerges from self-attention in diffusion models, which motivates the authors to decode instance boundaries from diffusion models to enable annotation-free instance and panoptic segmentation.
-	The paper is generally well-written and easy to follow.
-	The experiments are extensive and the results seem promising.

**Weaknesses:**

-	Some related works are missing. For example, [1, 2] are diffusion-based instance/semantic segmentation methods that also leverage the attention maps directly from diffusion models without relying on any label supervision. The authors should discuss these works as well.
-	The experimental settings mainly focus on the closed set. Given the open-set nature of diffusion models, I am curious about the effectiveness of the proposed method for open-vocabulary segmentation.
-	How to measure the quality of instance boundaries? Currently, it somewhat manifests through the downstream segmentation performances. Would it be possible to have a metric to directly evaluate instance boundaries before downstream training?
-	What are the failure cases of the proposed method? Adding more analysis on failure cases will add more insights to the paper.

**References:**

[1] MosaicFusion: Diffusion Models as Data Augmenters for Large Vocabulary Instance Segmentation. In IJCV, 2024.

[2] EmerDiff: Emerging Pixel-level Semantic Knowledge in Diffusion Models. In ICLR, 2024.

**Questions:**

See the questions mentioned above. Given the current status of the paper, I am leaning towards borderline accept and hope the authors could address my concerns during the rebuttal.

---

> ### Author Response · Authors · 2025-11-25
> **Response to Reviewer rCqE**
>
> We are grateful for Reviewer rCqE’s encouraging feedback, particularly for highlighting that our method is "well motivated" and the paper is "easy to follow." Your insightful suggestions regarding related works and additional evaluations have been instrumental in significantly strengthening our manuscript. Below, we address your comments point-by-point.
>
> **[W1] Discussion on Missing Related Works**
>
> Thank you for pointing out these relevant studies [1, 2]. We have added a detailed discussion in Appendix B.1. As you noted, while previous diffusion-based methods (e.g., DiffCut, DiffSeg, EmerDiff [2]) leverage attention for semantic segmentation, TRACE is distinct in its ability to decode instance-level boundaries early in the denoising process.
>
> Regarding MosaicFusion [1], it generates synthetic training data by stitching multiple single-instance images into a $2\times2$ mosaic. In contrast, TRACE discovers instance boundaries within a single natural image containing adjacent instances, handling complex occlusions and interactions that simple mosaic composition cannot model. This highlights the novelty of TRACE in extracting inherent instance cues from standard denoising, rather than relying on synthetic composition.
>
> **[W2] Extension to Open-Vocabulary Segmentation**
>
> We agree that the generative nature of diffusion models makes them naturally suitable for open-world scenarios. To validate this, we integrated TRACE into two state-of-the-art open-vocabulary segmentation models, TTD and Talk2DINO, and evaluated them across 5 standard benchmarks. As shown in **Table R1** (added as Table 13 in Appendix E.3), TRACE yields consistent improvements (e.g., +3.8 mIoU on VOC, +3.4 mIoU on Cityscapes).
>
> **Table R1. Effect of TRACE (Backbone: SD3.5-L) on open-vocabulary segmentation.**
> | Method | VOC | Context | Stuff | ADE | City |
> | :--- | :---: | :---: | :---: | :---: | :---: |
> | TTD | 61.1 | 37.4 | 23.7 | 17.0 | 27.9 |
> | + TRACE (Ours) | **64.9** | **40.3** | **24.3** | **18.3** | **31.3** |
> | Talk2DINO | 65.8 | 42.4 | 30.2 | 22.5 | 38.1 |
> | + TRACE (Ours) | **68.4** | **44.8** | **32.1** | **23.9** | **40.2** |
>
> These gains confirm that TRACE effectively refines the boundaries between adjacent classes and suppresses pixel bleeding in open-set scenarios, demonstrating its robustness beyond closed-set benchmarks.
>
> **[W3] Evaluation of Instance Boundary Quality**
>
> You raised an important point about evaluating our instance edges directly. Standard edge benchmarks (e.g., BSDS) focus on low-level cues like color or texture, which often result in false positives for instance segmentation. To address this, we constructed a ground-truth instance edge benchmark from the COCO 2014 validation set (see Fig. 11). We compared TRACE against four representative edge detectors using ODS/OIS (precision/recall) and clDice (topological connectivity), with metric details provided in Appendix F.
>
> **Table R2. Instance edge quality on COCO 2014.**
> | Method | ODS | OIS | clDice |
> | :--- | :---: | :---: | :---: |
> | Canny | 0.129 | 0.202 | 0.134 |
> | HED | 0.347 | 0.443 | 0.446 |
> | PiDiNet | 0.362 | 0.450 | 0.574 |
> | DiffusionEdge | 0.428 | 0.485 | 0.576 |
> | TRACE (Ours) | **0.889** | **0.899** | **0.826** |
>
> As shown in **Table R2** (i.e., Tab. 6), TRACE more than doubles the ODS/OIS of the best baseline (DiffusionEdge, 0.428) and achieves significantly higher topological connectivity (TRACE: 0.826 vs. Best Baseline: 0.576). This proves that the performance gains in downstream tasks stem directly from the superior quality of TRACE’s instance boundaries (structure) rather than low-level edges (texture).
>
> **[W4] Failure Case Analysis**
>
> To provide the balanced insight you requested, we have added a detailed analysis of limitations in Appendix E.4, supported by quantitative results (Tabs. 14/15) and qualitative visualizations (Fig. 15). We identified two primary failure modes:
>
> 1. Tiny Objects (Satellite Imagery): On datasets like HRSID where instances are extremely small ($\approx 0.01$\% of image area), the spatial compression of the VAE in latent diffusion models blurs boundaries, leading to a $\approx 5$\% drop compared to the baseline.
>
> 2. Domain Gap (Medical Imagery): On histopathology data (MoNuSeg), the natural-image priors of the diffusion backbone focus on irrelevant textures, reducing performance by $\approx 3$\%.
>
> These findings clarify the operating scope of TRACE and suggest that domain-specific fine-tuning is necessary for such specialized distributions.
>
> We hope these clarifications and additional experiments fully address your concerns. We are happy to provide further details if needed.

---

> > ### Comment · Reviewer_rCqE · 2025-11-27
> > **Official Comment by Reviewer rCqE**
> >
> > Thanks for the authors' detailed response. My concerns are well addressed. I would like to increase my score to 8.

---

> > > ### Author Response · Authors · 2025-11-27
> > >
> > > Thank you for your positive feedback and for increasing your score! We are delighted that our response and additional evaluations have satisfactorily addressed your concerns.
> > >
> > > Your constructive suggestions, particularly regarding the related works and open-vocabulary segmentation, have further enriched the quality and scope of our paper. We will ensure that all discussed revisions are faithfully incorporated into the final manuscript, and we will also release the updated codebase to support the full reproducibility of the rebuttal experiments.

---

### Official Review · Reviewer_inKu · 2025-11-01

**Soundness:** 2
**Presentation:** 2
**Contribution:** 2
**Rating:** 4
**Confidence:** 4

**Summary:**

This paper proposes TRACE (TRAnsforming diffusion Cues to instance Edges), a framework for decoding instance boundaries from the self-attention of pre-trained text-to-image diffusion models. It aims to solve the problem of difficult instance separation in unsupervised and weakly-supervised segmentation.

**Strengths:**

1.The idea that instance-level structural information (not just semantic information) is hidden in the self-attention (rather than cross-attention) of pre-trained diffusion models is interesting.
2. As a plug-and-play module, TRACE shows consistent and significant performance improvements when integrated into various existing segmentation frameworks.

**Weaknesses:**

1. It seems that the proposed method depends on specific diffusion model backbones. The discussion on detailed generalizability across model scales and architectures is required. Whether this dependency might limit its application in resource-constrained scenarios.
2. The paper compares TRACE with alternatives like Canny/HED/PiDiNet/Depth-based methods in tables/figures in Fig. 10, but it lacks the direct edge quality evaluation.

**Questions:**

1. Although the authors claim that the IEP timestep distribution is "model-agnostic," the results in Table 5 indicate that TRACE's final performance is strongly correlated with the chosen diffusion model backbone. For example, SD3.5-L (8.1B parameters) achieves an $AP^mk$ of 8.2, while SD1.5 (0.8B parameters) only reaches 6.8. The authors should discuss in more detail the method's generalizability across model scales and architectures, and whether this dependency might limit its application in resource-constrained scenarios.
2. In Section 3.4, it is ambiguous when describing the one-step self-distillation. The authors do not explicitly state how the "uncertain" pixels (value -1) are handled (or "excluded") within this loss function. This lack of critical implementation detail could hinder reproducibility. It does not sufficiently explain the specific impact of this threshold choice on the results (edge connectivity, false positives, false negatives), nor why $\mu+\sigma$ is robust across various scenarios. This step directly impacts the quality of the distillation supervision.
3. The paper compares TRACE with alternatives like Canny/HED/PiDiNet/Depth-based methods in tables/figures, showing TRACE's advantages. However, it lacks: a) Edge quality metrics (e.g., Precision/Recall/F1, ODS/OIS) on standard edge detection benchmarks (like BSDS, or COCO edge annotations if available); and b) Quantitative connectivity/topological metrics for the output edges. Currently, the quality of the edges is primarily demonstrated indirectly through downstream segmentation metrics.

---

> ### Author Response · Authors · 2025-11-25
> **[1/2] Response to Reviewer inKu**
>
> We deeply appreciate Reviewer inKu for the careful assessment and for recognizing the novelty of our core idea ("instance-level structure hidden in self-attention") and the "plug-and-play" effectiveness of TRACE. We acknowledge your concerns regarding generalizability and implementation details. Your constructive feedback has guided us to clarify our claims and perform direct quantitative evaluations, significantly strengthening our paper. Below, we address your concerns point-by-point.
>
> **[Q1 & W1] Clarification on "Model-Agnostic" and Generalizability across Backbones**
>
> You correctly pointed out that TRACE's performance scales with the diffusion backbone capacity (e.g., SD3.5-L > SD1.5). We clarify that our claim of "model-agnostic" refers to the applicability of TRACE across five diffusion architectures without modification, rather than implying identical performance across all diffusion backbones.
>
> **1. Universal Applicability**: Unlike prior diffusion-based segmentation methods (e.g., DiffCut) that rely on manually fixed layers or timesteps specific to U-Net, TRACE automatically identifies the Instance Emergence Point (IEP). As shown in Fig. 15, this optimal timestep $t^*$ consistently emerges in the same instance-aware range (30-60%) across 5 different backbones (both U-Net and DiT families) and 2 datasets. This confirms that the semantic-to-instance transition is a fundamental property of existing text-to-image diffusion models, independent of specific architectures.
>
> **2. Consistent Improvement:** In Tab. R1 (i.e., Tab. 5), TRACE consistently outperforms non-diffusion backbones. Remarkably, even the smallest diffusion model (PixArt-α, 0.6B) achieves 7.1 AP$^mk$, which corresponds to a 2.3$\times$ improvement over the baseline (ProMerge, 3.1 AP$^mk$). This result is particularly striking because it significantly surpasses the largest non-diffusion foundation model, Qwen2.5-VL (72B), which only reaches 4.1 AP$^mk$ despite having 120× more parameters. This validates that the instance-aware cues are specific to the generative diffusion prior rather than general model capacity.
>
> **3. Scalability:** The performance gain from larger backbones (SD1.5 $\rightarrow$ SD3.5-L) reflects TRACE's ability to leverage richer generative priors. This is a desirable trait, as TRACE naturally benefits from the rapid advancements in foundation text-to-image diffusion models.
>
> **Table R1. Effect of backbone choice on COCO 2014.**
> | Method | Backbone | Params. | $AP^{mk}$ | $AR^{mk}_{100}$ |
> | :--- | :--- | :---: | :---: | :---: |
> | ProMerge | -- | -- | 3.1 | 7.6 |
> | | **_Non-diffusion_** | | | |
> | + TRACE | DINOv2-G | 1.1B | 2.6 | 7.7 |
> | + TRACE | EVA02-E | 5.0B | 3.2 | 7.9 |
> | + TRACE | DINOv3 | 7.0B | 4.3 | 8.9 |
> | + TRACE | LLaVA | 13B | 3.8 | 8.4 |
> | + TRACE | Qwen2.5-VL | 72B | 4.1 | 8.5 |
> | | **_Diffusion_** | | | |
> | + TRACE | SD1.5 | 0.8B | 6.8 | 11.2 |
> | + TRACE | PixArt-$\alpha$ | **0.6B** | 7.1 | 11.8 |
> | + TRACE | SDXL | 2.5B | 7.4 | 12.3 |
> | + TRACE | SD3.5-L | 8.1B | 8.2 | 13.1 |
> | + TRACE | FLUX.1 | 12B | **8.3** | **13.4** |
>
> **[Q2] Handling Uncertain Pixels in Self-Distillation**
>
> Thank you for highlighting the ambiguity. We have revised Sec. 3.4 and Algorithm 2 to explicitly state that uncertain pixels are excluded from the Dice loss computation via a binary mask $W_{i,j}$ (Eq. 10). This follows standard pseudo-labeling strategies to filter out ambiguous attention signals that introduce label noise.
>
> To validate this design, we compared our $\mu \pm \sigma$ strategy against a $\mu$-only baseline (treating all pixels as certain). As shown in Table R2 (added as Appendix E.1), excluding uncertain pixels is critical:
>
> **Table R2. Ablation on pseudo-labeling schemes for ABDiv.**
> | Method | Pseudo-Labeling | $AP^{mk}$ | ODS-Precision | ODS-Recall | ODS |
> | :--- | :---: | :---: | :---: | :---: | :---: |
> | ProMerge | -- | 3.1 | -- | -- | -- |
> | + TRACE | $\mu$ | 6.4 | 0.572 | 0.963 | 0.717 |
> | + TRACE | $\mu \pm \sigma$ | 8.2 | 0.852 | 0.950 | 0.889 |
>
> This strategy significantly boosts ODS-Precision (0.572 $\rightarrow$ 0.852) and downstream performance (6.4 $\rightarrow$ 8.2 AP), confirming that suppressing false positives from ambiguous regions is essential for training a robust edge decoder.
>
> (Continued in Next Response)

---

> ### Author Response · Authors · 2025-11-25
> **[2/2] Response to Reviewer inKu**
>
> (Continued from Previous Response)
>
> **[Q3 & W2] Evaluation of Instance Boundary Quality**
>
> We agree that evaluating instance edges only via downstream segmentation is indirect. To address this, we constructed a ground-truth instance edge benchmark from the COCO validation set (see Fig. 11) and compared TRACE directly against four representative edge detectors. Following your suggestion, we report standard edge metrics (ODS, OIS) and a topological connectivity metric (clDice), with metric details provided in Appendix F.
>
> **Table R3. Quantitative evaluation of instance edge quality.**
> | Method | ODS | OIS | clDice |
> | :--- | :---: | :---: | :---: |
> | Canny | 0.129 | 0.202 | 0.134 |
> | HED | 0.347 | 0.443 | 0.446 |
> | PiDiNet | 0.362 | 0.450 | 0.574 |
> | DiffusionEdge | 0.428 | 0.485 | 0.576 |
> | TRACE (Ours) | **0.889** | **0.899** | **0.826** |
>
> **Table R3** (i.e., Tab. 6) shows that TRACE achieves an ODS of 0.889, more than doubling the performance of the best baseline (DiffusionEdge, 0.428). As shown in Fig. 10, conventional edge detectors focus on low-level color/texture contrast, leading to many false positives within objects/instances. TRACE, driven by the hidden diffusion prior, specifically targets instance-level boundaries. Furthermore, the high clDice score (TRACE: 0.826 vs. Best Baseline: 0.576) confirms that TRACE produces topologically connected instance boundaries, which is critical for effective instance separation in downstream segmentation tasks (as evidenced by robust improvements across 11 real-world benchmarks in Tabs. 1/2/12/13).
>
> We hope these clarifications and additional rigorous evaluations fully address your concerns. We are confident that your constructive feedback has significantly improved the reliability and completeness of our work.

---

### Author Response · Authors · 2025-11-25
**Summary of Revisions**

Dear Reviewers,

We sincerely thank all reviewers for their insightful comments and constructive feedback. We are greatly encouraged that the reviewers recognized the **novelty** of utilizing diffusion self-attention for instance edge detection (Reviewers JkD6, KrZ9, rCqE, inKu), the **elegant and simple design** of TRACE (Reviewer KrZ9, JkD6, rCqE), and the **extensive ablations** and **strong empirical results** (Reviewers KrZ9, JkD6, rCqE).

In response to your valuable suggestions, we have enriched our manuscript with additional experiments and clarifications. Please note that we have merged the main text (9-to-10 pages) and the appendix (previously separate in ```Supplementary Material.zip```) into a single PDF file for this revised version. All changes are highlighted in **blue** for easy reference. Below, we summarize the key updates:

**1. Extended Experiments and Generalization**

To demonstrate the robustness of TRACE beyond the initial settings, we have added experimental results:

* **[Appendix E.3] Generalization to Diverse Domains:** We extended the evaluation to 7 UIS benchmarks and 5 open-vocabulary benchmarks.

* **[Discussion & Appendix F] Direct Evaluation of Edge Quality:** We constructed a ground-truth instance edge benchmark on COCO and quantitatively verified that TRACE achieves superior boundary precision (ODS/OIS) and topological connectivity (clDice) compared to conventional edge detectors.

* **[Discussion & Appendix E.3] Superiority of Generative Priors:** We compared TRACE across 10 different backbones (5 diffusion vs. 5 non-diffusion).

**2. In-depth Analysis and Theoretical Grounding**

We have expanded the analysis of our core components to address theoretical and implementation queries:

* **[Appendix E.2] Consistency of IEP:** We provided histogram analyses showing that the Instance Emergence Point (IEP) is robust across different datasets and model architectures.

* **[Appendix D] Theoretical View of IEP:** We added a lightweight information-theoretic derivation connecting the IEP to the peak of Fisher Information with respect to the noise level, providing a solid theoretical basis for our KL-based criterion.

* **[Appendix E.1] Ablation on Design Choices:** We clarified the impact of uncertainty masking and edge decoder capacity, confirming the efficiency of our lightweight design.

**3. Clarifications and Limitations**

* **[Appendix E.4] Failure Case Analysis:** We explicitly analyzed limitations in specialized domains (Satellite and Medical imagery) to clarify the operating scope of TRACE and suggest future directions for domain-specific adaptation.

* **[Appendix B] Expanded Related Work:** We included a detailed discussion on recent diffusion-based segmentation methods and data synthesis approaches to better position our contribution.

We trust that the revised manuscript and our responses have effectively resolved your concerns. We thank you once again for your dedication to reviewing our work, and we look forward to your comments on the revised version.

Sincerely,

Authors of Submission 11242

---

### Author Response · Authors · 2025-12-02
**Rebuttal Summary: Key Updates and Reviewer Ratings**

Dear Area Chair,

In light of the recent notification regarding the OpenReview security incident, we affirm full compliance with the ICLR Code of Conduct. We confirm that we have not communicated with any reviewer outside of OpenReview, and that all reviewer updates were made independently, based solely on the technical improvements and clarifications provided in our rebuttal. With this understanding, we respectfully request your consideration of the following summary of our rebuttal and reviewer engagement.

**1. Summary of Reviewer Ratings**

We are encouraged that our revision has led to a score increase and positive confirmation.

| Reviewer | Pre-Rebuttal Score | Post-Rebuttal Score | Note |
| :--- | :---: | :---: | :--- |
| Reviewer rCqE | 6 | **8 (Raised)** | Explicitly raised score; concerns fully addressed. |
| Reviewer KrZ9 | 8 | 8 (No response) | **Initial Accept.** Provided requested theoretical grounding & consistency checks. |
| Reviewer JkD6 | 6 | 6 (Confirmed) | Confirmed concerns on experimental details were addressed. |
| Reviewer inKu | 4 | 4 (No response) | Fully addressed generalizability/edge quality concerns with new extensive data. |

**2. Key Resolutions by Reviewer**

* **Reviewer rCqE (Score 6 $\rightarrow$ 8):** Explicitly raised the score to **Accept**, noting that "My concerns are well addressed." We provided four key results: 1) Clarified the novelty of our instance cues versus diffusion-based semantic cues, 2) Extended evaluations to open-vocabulary segmentation (Table 13), 3) Verified quantitative instance edge quality, and 4) Analyzed failure cases.

* **Reviewer KrZ9 (Score 8): Initial Accept.** Although there was no further response due to the system freeze, we reinforced the grounds for the positive assessment by providing the requested theoretical derivation for IEP (Appendix D), verifying the consistency of $t^*$ (across seeds, datasets, and classes; Figure 14), and analyzing limitations in two unseen domains.

* **Reviewer JkD6 (Score 6):** Confirmed that our response "addresses concerns on the experimental details." We validated the method's robustness regarding text prompt dependency and edge decoder capacity through additional ablation studies (Table 10).

* **Reviewer inKu (Score 4):** We addressed concerns about generalizability by verifying stability and scalability across 10 backbones (5 diffusion vs. 5 non-diffusion, ranging from 0.6B to 72B; Table 5). Additionally, we provided a quantitative benchmark showing $>2\times$ improvement in instance edge quality compared to previous SOTA (Table 6).

We trust that this summary, alongside the revised manuscript and the reviewers' positive feedback, clearly demonstrates that all concerns have been resolved. To contribute to the research community and ensure full reproducibility, we are committed to releasing our complete codebase, including all scripts for the new benchmarks and ablation studies conducted during this rebuttal. We sincerely appreciate your dedication to managing our submission under these exceptional circumstances.

Best regards,

Authors of Submission 11242

---

### Public Comment · ~Venkatesh_Thirugnana_Sambandham1 · 2026-03-01
**Project page not accessible**

The project page returns a 404

---

> ### Public Comment · ~Sanghyun_Jo1 · 2026-03-02
>
> Thank you for pointing this out. We have updated the link, and we plan to upload additional materials (e.g., a video and checkpoints) in April. We appreciate your interest and patience.

---

### Meta-Review · Area_Chair_ZQn3 · 2026-01-05

**Summary:**

TRACE introduces a novel unsupervised framework identifying instance edges within diffusion model self-attention via the Instance Emergence Point (IEP). Reviewers praised the original insight, computational efficiency, and strong empirical performance. The authors provided a comprehensive rebuttal including theoretical grounding for IEP and new edge benchmarks, securing a consensus for acceptance.

**Reviewer Concerns:**

Primary concerns were regarding generalizability across architectures, theoretical justification for  IEP, and a lack of direct edge quality metrics. The rebuttal resolved these by benchmarking diverse backbones, providing a Fisher Information derivation for IEP, and adding standard edge metrics (ODS/OIS). No major concerns remain outstanding.

**Reviewer Scores:**

Reviewers rCqE (6 ->8) , KrZ9 (8) and JkD6 (6) have given high positive score. Reviewer inKu (4) had concerns on generalizability, which was fully addressed by the new backbone comparisons.

---

### Decision · Program_Chairs · 2026-01-26

Accept (Oral)